**Data Availability Statement:** The data are within the paper.

**Funding:** The authors received no specific funding for this work.

# Data-driven models for atmospheric air temperature forecasting at a continental climate region

**Mohamed Khalid Alomar**[1], **Faidhalrahman Khaleel**[1], **Mustafa M. Aljumaily**[1], **Adil Masood**[2], **Siti Fatin Mohd Razali**[3], **Mohammed Abdulhakim AlSaadi**[4], **Nadhir Al-Ansari**[5]*, **Mohammed Majeed Hameed**[1,3]*

1 Department of Civil Engineering, Al-Maarif University College, Ramadi, Iraq, 2 Department of Civil Engineering, Jamia Millia Islamia, New Delhi, India, 3 Department of Civil Engineering, Faculty of Engineering and Built Environment, Universiti Kebangsaan Malaysia, Selangor, Malaysia, 4 Natural and Medical Sciences Research Center, University of Nizwa, Nizwa, Oman, 5 Civil Engineering Department, Environmental and Natural Resources Engineering, Lulea University of Technology, Lulea, Sweden

* m.majeed@uoa.edu.iq, mohmmag1@gmail.com (MMH); nadhir.alansari@ltu.se (NAA)

## Abstract

Atmospheric air temperature is the most crucial metrological parameter. Despite its influence on multiple fields such as hydrology, the environment, irrigation, and agriculture, this parameter describes climate change and global warming quite well. Thus, accurate and timely air temperature forecasting is essential because it provides more important information that can be relied on for future planning. In this study, four Data-Driven Approaches, Support Vector Regression (SVR), Regression Tree (RT), Quantile Regression Tree (QRT), ARIMA, Random Forest (RF), and Gradient Boosting Regression (GBR), have been applied to forecast short-, and mid-term air temperature (daily, and weekly) over North America under continental climatic conditions. The time-series data is relatively long (2000 to 2021), 70% of the data are used for model calibration (2000 to 2015), and the rest are used for validation. The autocorrelation and partial autocorrelation functions have been used to select the best input combination for the forecasting models. The quality of predicting models is evaluated using several statistical measures and graphical comparisons. For daily scale, the SVR has generated more accurate estimates than other models, Root Mean Square Error (RMSE = 3.592°C), Correlation Coefficient (R = 0.964), Mean Absolute Error (MAE = 2.745°C), and Thiels' U-statistics (U = 0.127). Besides, the study found that both RT and SVR performed very well in predicting weekly temperature. This study discovered that the duration of the employed data and its dispersion and volatility from month to month substantially influence the predictive models' efficacy. Furthermore, the second scenario is conducted using the randomization method to divide the data into training and testing phases. The study found the performance of the models in the second scenario to be much better than the first one, indicating that climate change affects the temperature pattern of the studied station. The findings offered technical support for generating high-resolution daily and weekly temperature forecasts using Data-Driven Methodologies.

**Competing interests:** The authors declare that there are no conflicts of interest.

## 1. Introduction

It is well-known that numerous meteorological and ecological events, human life, and crops in agricultural areas are significantly influenced by climate conditions as well as several factors related to the environment's physical conditions. The natural resources that provide humans basic needs and opportunities for social and economic development are part of the physical environment, including land, air, and water. A clean and healthy environment is one of the essential principles that should be preserved and protected [1]. The temperature parameter is seen as the most influential parameter out of all meteorological parameters, which reflects the effect of climate change on earth and its surrounding atmosphere. Recently, climate change has caused extreme natural phenomena such as heat waves, severe winters, heavy snowfall, and droughts worldwide, leading to environmental and health crises [2–6]. Air temperature prediction helps meteorologists to know the likelihood of hurricanes and floods in an area [7].

Various meteorological parameters such as rainfall, humidity, atmospheric pressure, wind speed, solar energy, and soil temperature are significantly correlated to air temperature [8]. Moreover, air temperature is one of the most influential factors in evapotranspiration, which is vital for managing water resources and agricultural activities [9]. Accurate air temperature prediction is substantial in many decision-making sectors, such as energy, agriculture, transportation, and tourism [10]. Additionally, accurately predicting air temperature is the most crucial aspect of environmental studies involving operational eco-environmental systems. From the industrial aspect, predicting air temperature is essential in energy management strategies to obtain comfortable indoor temperatures and eventually reduce the consumption of energy [11].

According to the literature, two main approaches have been used to predict air temperature: general circulation (GCM) and statistical models. The GCM is utilized to comprehend the dynamics behind climate system physical components, derive global temporal and spatial changes and make predictions based on the future forcing of greenhouse gasses and aerosols [12]. GCMs can be applied to the problem of attributing climate change from a season to a decade ahead. Conversely, statistical models attempt to determine whether climate change is externally driven by minimizing the utilization of complex climate models. They are generally more straightforward and less computationally intensive than the GCMs, and several studies have showed that the use of statistical models has produced results consistent with GCMs. Various statistically based approaches have been proposed recently, several of which have been developed in the econometric literature. The statistical models can be categorized into two approaches: cointegration approaches which determine the relationship between non-stationary and stationary times series [13], and regression approaches which evaluate the characteristics of time series for a given temperature data [14, 15]. However, since temperature prediction involves high nonlinearity and dimensionality, the statistical models faced some drawbacks in capturing them [16].

Meanwhile, machine learning (ML) approaches have attracted much attention due to their superlative performance in dealing with high nonlinearity phenomena [17, 18] and solving complex problems such as drought [19–24], rainfall [25–29], evapotranspiration [30–34] and streamflow [35–38]. For example, a study was conducted in the Queensland area where ML models' performances were compared with the Australian Predicted Ocean-Atmosphere Model (POAMA) for precipitation prediction. The POAMA model showed a significant improvement in the predictive performance of the ML modeling framework. It was reported that the performance of the neural network (NN) model was superior to POAMA in precipitation prediction over three regions in Queensland [39–41]. For temperature prediction, A. Sekertekin et al. [6] used the adaptive neuro-fuzzy inference system (ANFIS) and long-short

term memory (LSTM) network to predict temperature for both ultra short-term and short term period(hourly and one day ahead). The results showed that the LSTM model was able to efficiently predict the temperature for both the time scales. However, the LSTM has several disadvantages, such as it requires longer time and more memory to train. Besides, its parameters are difficult to assign and implement and the outcomes are vulnerable to various random weight initializations. S. Salcedo-Sanz et al. [12] used the support vector regression (SVR) and multi-layer perceptron (MLP) models to predict the mean monthly air temperature. The dataset from the monitoring stations located in New Zealand and Australia was used for the model development. The results showed that the SVR model provided the best accuracy in temperature prediction. Overall, very few studies based on daily temperature prediction have been conducted for regions with continental climatic conditions Therefore, the main objective of this study is to forecast air temperature over a continental climate case study which is in North America. Two-time scales are adopted in this study, daily and weekly. For fulfilling this task, four Data-driven models i.e., Support Vector Regression (SVR), Regression Tree (RT), Quantile Regression Tree (QRT), and Gradient Boosting Regression (GBR) have been applied. These models have been used to predict one-day and one-week temperature ahead depending on the past temperature values for both time scales (weekly and daily). Comprehensive comparisons supported by statistical measures and comparative figures have been applied to select the most efficient models.

## 2. Methodology

### 2.1 Case study

North Dakota is located in the middle of North America and is subjected to extreme climate conditions, with hot summers and cold winters. Due to its inland location and proximity to both the North Pole and the Equator, which are almost equal, there are noticeable temperature fluctuations. Furthermore, it has been observed that the temperature varies extremely from season to season, which may be responsible for the changes in weather throughout the time [42]. Since North Dakota has a continental climate, forecasting the patterns of meteorological parameters is a challenging task. The difficulty in simulating weather parameters in such region may be due to the nature of the fluctuating climate during the seasons.

Tables 1 and 2 show the statistical characteristics of the minimum, mean, average, standard deviation, and skewness of the daily and weekly air temperature values at the Crary

**Table 1. Statistical characteristics of Crary station: Daily scale.**

| Month/statics | Min | Average | Max | St. D | Skewness |
|---|---|---|---|---|---|
| Jan | -35.50 | -13.018 | 4.12 | 7.905 | -0.194 |
| Feb | -32.55 | -12.847 | 5.70 | 7.695 | 0.015 |
| Mar | -28.19 | -4.191 | 14.85 | 7.465 | -0.349 |
| Apr | -11.80 | 4.542 | 18.38 | 5.919 | -0.320 |
| May | -1.67 | 11.860 | 24.87 | 5.072 | -0.008 |
| Jun | 5.15 | 17.944 | 30.56 | 3.626 | -0.101 |
| Jul | 12.28 | 20.799 | 27.65 | 2.916 | -0.340 |
| Aug | 9.19 | 19.408 | 28.28 | 3.248 | 0.147 |
| Sep | 0.66 | 14.598 | 27.89 | 4.555 | -0.078 |
| Oct | -8.15 | 6.098 | 22.86 | 5.436 | 0.137 |
| Nov | -19.79 | -2.479 | 12.81 | 6.430 | -0.165 |
| Dec | -30.83 | -10.901 | 5.00 | 7.147 | -0.320 |

**Table 2. Statistical characteristics of Crary station: Weekly scale.**

| Month/statics | Min | Average | Max | St. D | Skewness |
|---|---|---|---|---|---|
| Jan | -28.23 | -13.112 | -1.84 | 5.882 | -0.070 |
| Feb | -27.35 | -13.191 | 0.26 | 6.093 | 0.111 |
| Mar | -20.56 | -5.287 | 11.32 | 6.311 | -0.115 |
| Apr | -10.33 | 3.833 | 12.74 | 4.926 | -0.383 |
| May | 3.22 | 11.185 | 21.55 | 3.879 | 0.101 |
| Jun | 10.28 | 17.461 | 23.22 | 2.743 | -0.133 |
| Jul | 15.76 | 20.644 | 24.47 | 1.949 | -0.384 |
| Aug | 13.98 | 19.650 | 25.50 | 2.292 | -0.056 |
| Sep | 6.20 | 15.149 | 22.96 | 3.319 | -0.229 |
| Oct | -3.89 | 7.212 | 18.09 | 4.414 | 0.036 |
| Nov | -14.03 | -1.822 | 8.59 | 5.210 | -0.133 |
| Dec | -22.77 | -9.993 | 1.34 | 5.816 | -0.380 |

meteorological station from 2000 to 2021. According to reported data, the three hottest months are June (17.944 C°), July (20.799 C°), and August (19.408 C°) while the coldest months in this case study are December (-10.901 C°), January (-13.018 C°), and February (-12.847 C°). Furthermore, the recorded air temperatures have extreme values (far from the mean) in four months (i.e., December to March). In these months, the standard deviation values of the data are very high compared to other months. Nevertheless, the dispersion of the data through June (St. D = 3.626), July (St. D = 2.916), and August (St. D = 3.248) is very little, which means that the data are more consistent with their normal rates. Notably, the utilized data in this study are collected from the open-source website of the Crary station [43]. Finally, the location of the studied region is shown in Fig 1.

## 2.2 Support vector regression

Support vector regression (SVR) is considered a powerful and efficient tool based on the notion of statistical learning and was first introduced by Vapnik [44] to describe regression as a part of the support vector machine (SVM). Based on the principle of structural risk minimization (SRM), SVR has been successfully implemented in real-world challenge modeling by overcoming classification and regression tasks [45]. The linear relationship between independent variables ($x_1, x_2, x_3, \cdots, x_r$) and dependent variable (y) is given in the equation below.

$$y = f(x) = w\emptyset(x) + b \tag{1}$$

Where $w_i$ and $b$ are the weight and bias of the model, respectively. $\emptyset(x)$ is the higher dimensional feature space converted from the independent vector (input). These parameters can be determined by minimizing $\|w\|^2 = (w.w)$ as follows

$$\min \frac{1}{2} \| w^2 \| + C \sum_{i=1}^{r} (\xi_i + \xi_i^*) \tag{2}$$

$$\text{under the constraints} \begin{cases} y_i - w\emptyset(x) - b \leq \varepsilon + \xi_i \\ y_i - w\emptyset(x) - b \geq -\varepsilon - \xi_i^* \, \forall_i \in \{1, \ldots n\} \\ \xi_i, \xi_i^* \geq 0 \end{cases} \tag{3}$$

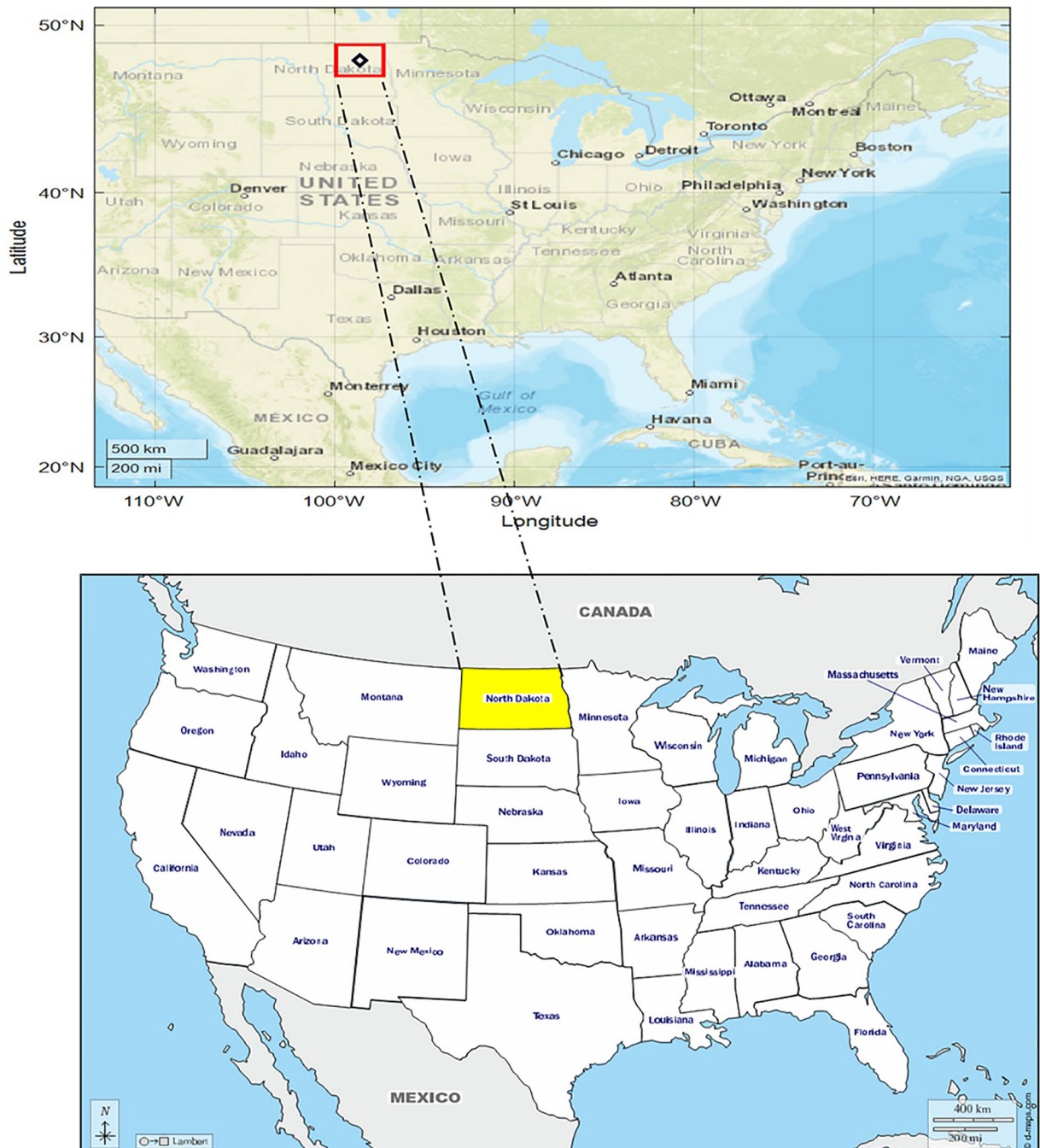

**Fig 1. The location of Crary station.** Source (https://d-maps.com/index.php?lang=en; https://www.usgs.gov/).

Where, C is the regularization constant, $\xi_i$, $\xi_i^*$ are the slack variables and $\varepsilon$ is the size of the tube, "denoting the accuracy of the function to be approximated" [46].

Based on Lagrange multipliers, the standard SVR can overcome the following optimization problem.

$$min -\frac{1}{2}\sum_{ij}^{r}(\alpha_i - \alpha_i^*)(\alpha_j - \alpha_j^*)(x_i, x_j) - \sum_{i=1}^{r}(\alpha_i + \alpha_i^*) + \sum_{i=1}^{r}\gamma_i(\alpha_i - \alpha_i^*) = 0 \quad (4)$$

Which is subjected to:

$$\sum_{i=1}^{r} \left( \alpha_i - \alpha_i^* \right) = 0 \qquad (5)$$

$$\alpha_i, \alpha_i^* \in [0, \rho] \qquad (6)$$

Where $\rho$ are the cost factor, $\alpha_i, \alpha_i^* \geq 0$ are the Lagrange multiplier factors. The linear SVR can be written as follows

$$f(x) = \sum_{i=1}^{r} \left( \alpha_i - \alpha_i^* \right)(x_i, x) + b \qquad (7)$$

This equation may be considered inappropriate for solving many engineering problems because of its linear characteristic, while engineering problems often need a non-linear regression analysis. Therefore, in order to switch the input data to a much higher-dimensional space, nonlinear kernel functions are utilized. In this regard, the radial bases kernel function (*RBF*) is used in this study and can be expressed mathematically in the equation below.

$$K(x_i, x) = \exp(-\beta \parallel x - x_i \parallel^2) \qquad (8)$$

Where $K(x_i, x)$ represents the kernel function and $\beta$ is the bandwidth of $K(x_i, x)$.

## 2.3 Regression tree and quantile regression tree

A decision tree (DT) is a supervised machine learning-based technique that uses labeled data (data with known target attributes) to carry out simulations with the help of classification and regression algorithms [47]. In general, DT's consist of three types of nodes: decision Root nodes, internal nodes, and leaf nodes, where each node or leaf denotes a class label while the branches denote the outcome of the test performed [48]. The technique splits the input dataset on the basis of the most significant splitter or differentiator in the input variables. This process of data division and selection of the most significant attribute in the dataset is governed by the classification and regression algorithms. The technique follows a top-down approach as the top portion holds all the observations at one spot, which splits into two or more branches that further split. This approach is also referred to as the greedy approach, as it only incorporates the current nodes without focusing on the future nodes [49]. The decision tree algorithm continues to run until a stop criterion such as the minimum number of observations etc., is attained. Once this criterion is achieved and a decision tree is developed, many nodes are detected as outliers which may be addressed through the tree pruning method. This, in turn, improves the forecasting accuracy of the DT-based model.

In the same method that regression minimizes cost function (i.e., squared-error loss) when forecasting a single point estimate, the quantile regression tree (QRT) minimizes the loss function in forecasting a particular quantile. The median, or 50th percentile, is the most commonly used quantile, and the quantile loss is just the sum of absolute errors in this case. Additionally, quantiles can be used as endpoints of prediction intervals; for instance, the 10th and 90th percentiles define an 80-percentile range in the middle. It appears that the quantile loss differs according to the evaluated quantile, with higher quantiles penalizing more for negative errors and lower quantiles penalizing more for positive errors. Accordingly, in this study, we used the median (the 50th percentile), which is the most well-known quantile.

In the fields of artificial intelligence and search algorithms, pruning is a data compression method used to minimize the size of decision trees by deleting parts of the tree that are deemed

non-critical and repetitive to the regression of instances. By assessing the predictive value of each node in a regression tree, regression tree pruning decreases the danger of overfitting. Nodes that do not increase the anticipated prediction efficiency on new data are substituted with leaves.

## 2.4 Gradient Boosting Regression (GBR)

GBR is an ensemble machine learning approach that enhances the prediction performance of a classical decision tree by incorporating a sequential statistical process called boosting, of which the principle idea is to combine a set of weak predictive models to form a single and high accurate predictive model [50, 51]. The technique applies an iterative procedure, where the estimates of the new tree model (weak learner) are updated with the pseudo residuals (negative gradient of the loss function) of the current learner [52]. This process is repeated until the loss function of the model is reduced to a minimum value, thus improving the forecasting performance of the model.

The iterative training process of the GBRT with K decision trees can be briefly explained as follows:

For a given training dataset D = {$(x_1, y_1), (x_2, y_2), \ldots, (x_n, y_n)$}, the loss function is computed as:

$$L(y, f(x)) = (y - f(x))^2 \tag{9}$$

Step 1. Initialize the new tree model (weak learner) with a constant value:

$$f_0(x) = arg \min_c \sum_{i=1}^{N} L(y_i, c) \tag{10}$$

Step 2. Assume the number of iterations $m = 1, 2, 3. \ldots, K$
(a) For $i = 1,2,3. \ldots, N$. The pseudo residuals of the $i_{th}$ training data is calculated as:

$$r_{mi} = -\left[\frac{\partial L(y, f(x_i))}{\partial f(x_i)}\right]_{f(x) = f_{m-1}(x)} \tag{11}$$

(b) Fit a regression tree in terms of $r_{mi}$, and deduce the leaf node area $R_{ml}$ of the $m_{th}$ tree. Predict the leaf node area of the decision tree to attain an approximate value of the fitting residual.

(c) For $l = 1,2,3. \ldots, L$. Adopt linear search to attain the value in the leaf node range and minimize the loss function with gradient descent. The best residual fitting value of each blade is as follows:

$$c_m = arg \min_c \sum_{i=1}^{N} L(y_i, f_{m-1}(x_i) + c) \tag{12}$$

(d) Update the regression tree

$$f_m(x) = f_{m-1}(x) + \sum_{l=1}^{L} c_{ml} I(x \in R_{ml}) \tag{13}$$

Step 3. Obtain the final model

$$f(x) = f_M(x) = \sum_{m=1}^{M} \sum_{l=1}^{L} c_{ml} I(x \in R_{ml}) \tag{14}$$

## 2.5 Autoregressive model

The autoregressive integrated moving average (ARIMA) is a historical data-based model and is considered the most common time series modeling approach first introduced by Box and Jenkin in 1976 [53]. ARIMA model is considered a hybrid model in which the Autoregressive (AR) and moving average (MA) models generalized forms are combined for modeling non-stationary univariate time series data by approximating the time series using a mathematical model based on past and current values. The utilization of ARIMA model is done by setting the order of three terms: autoregressive, sequence difference, and moving average. The general successive difference equation for d$^{th}$ order can be mathematically expressed as follows.

$$\Delta^d T_t = (1 - B)^d T_t \tag{15}$$

Where $d$ is the order of sequence difference and $B$ is the backshift operator. The general *ARIMA* equation can be briefly presented as follows [54].

$$\emptyset_p(B)w_t = \theta_q(B)e_t \tag{16}$$

Where $\emptyset_p(B)$ is the autoregressive operator of order $p$, $\theta_q(B)$ is the moving average term of order $q$ and $w_t = \Delta T_t$.

## 2.6 Random forest

Random forest is a supervised machine learning technique which is made up of large number of small decision trees, known as estimators, which generate their own predictions. 'Forest' generated by the random forest algorithm is trained through bagging or bootstrap aggregating [55]. Bagging is an ensemble meta-algorithm that fine-tunes the prediction accuracy of machine learning algorithms. The (random forest) algorithm produces the output based on the predictions of the decision trees. It predicts by taking the average or mean of the output from various trees. Increasing the number of trees increases the precision of the outcome. The various advantages of this technique over other machine learning approaches such as need of less computation time, ease of working with high-dimensional data, strong fault tolerance and parallel processing make it suitable even for very high-dimensional problems like air temperature forecasting.

## 2.7 Model development

In this work, four regression models i.e., SVR, GBR, QRT, and RT have been used to predict the daily and weekly air temperatures over the continental climatic region of North America. The time-series data was collected from the Crary meteorological station from 2000 to 2021. For selecting the best input lags, the autocorrelation function (ACF) and partial autocorrelation function (PACF) have been used to analyze the data. According to Fig 2, the ACF provides more information on the time series properties like stationary, trend pattern, seasonality, and randomness. The daily and weekly temperature patterns were examined to determine the most appropriate predictors utilizing correlation statistics such as ACF and PACF, respectively. The statistical techniques used the time-lagged data from temperature time series to estimate the daily and weekly intervals between the present T value and prior T value for any given observation (i.e., a time lag) [55]. Thus, selecting which lags have a significant correlation and significant information may benefit. Besides, the lags confined between upper and lower bounds are neglected because they have lower correlations and represent the white noise in the time series, which cannot be predicted. Both ACF and PACF are provided in the following

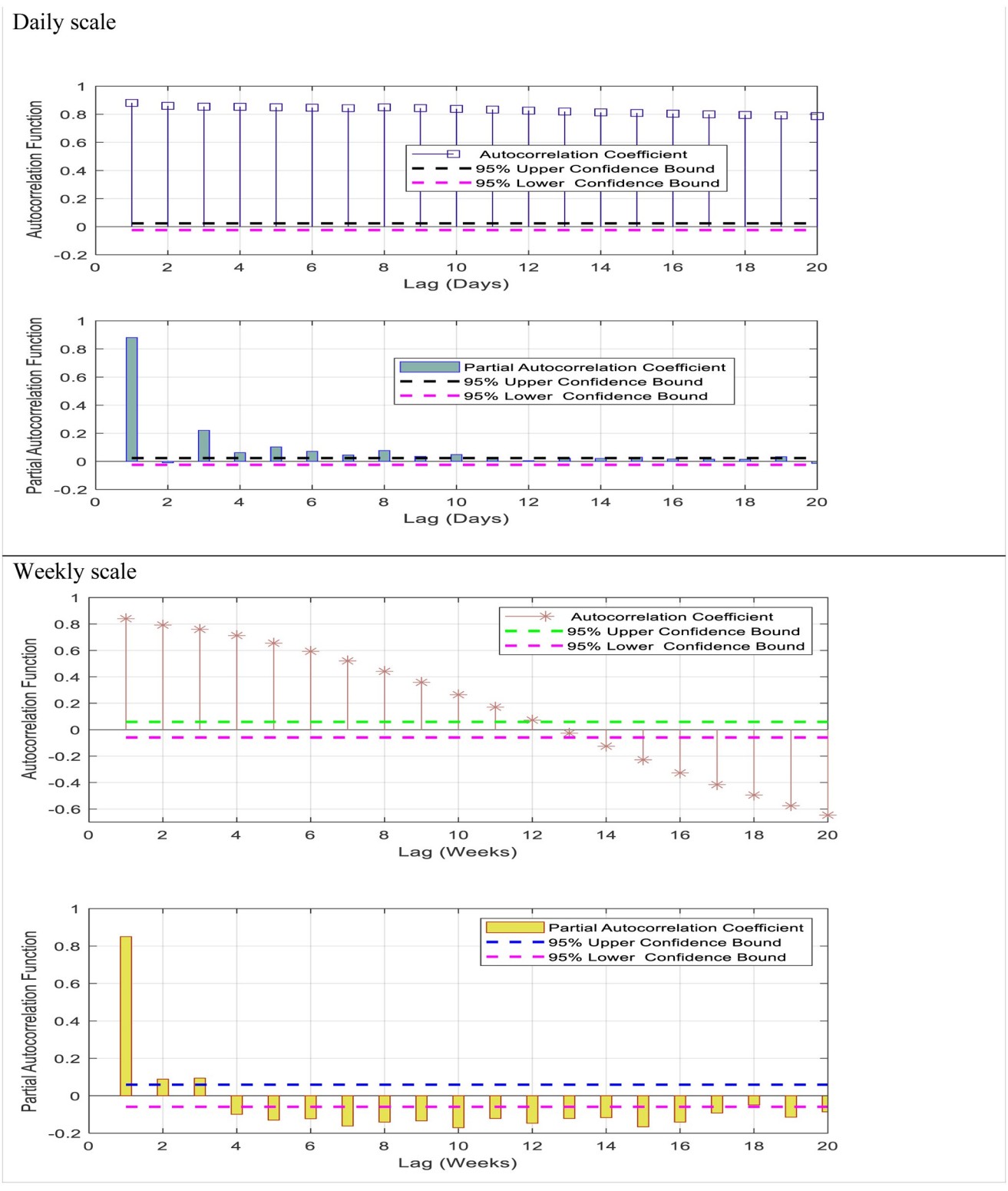

**Fig 2. ACF and PACF for input determination.**

equations:

$$\text{ACF} = \frac{\sum_{t=1}^{N-k}(X_t - \bar{x})(X_{t+k} - \bar{X})}{\sum_{t=1}^{N-k}(X_t - \bar{X})} \tag{17}$$

$$\text{PACF} = \frac{\text{ACF} - \sum_{j=1}^{k-1}\text{PACF}_{k-1,j}\text{ACF}_{k-1}}{1 - \sum_{j=1}^{k-1}\text{PACF}_{k-1,j}\text{ACF}_{k-1}} \tag{18}$$

The N is the total number of temperature records, $X_t$ and $-X$ is the time series record at time t and the mean of the temperature records, and finally, the k is number of lags in the time series data. lower and upper limits (UP, LO) can be determined at 95% significant level by the following equation:

$$\text{LO, UP} = \pm\frac{1.96}{\sqrt{N}} \tag{19}$$

The ACF declines more slowly concerning the daily scale, which means that the time series data is not stationary. It is challenging to select the most effective lags for daily scale using ACF because of seasonality. The PACF, similar to the ACF, shows the association between two records that the shorter delays between those observations do not describe. For instance, the partial autocorrelation coefficient for the third lag in the daily scale temperature is only a correlation that the previous short lags (lag two and lag one) have not explicitly explained. Therefore, the PACF is more suitable for selecting the input lags for predicting the short scale of time series than the ACF. Table 3 shows the input combinations used in this study. It is worth mentioning that 70% of the data is used to train the suggested models, and 30% of the data included the end of the time series of the data utilized for the testing phase and checking the models' performances (see Fig 3). The following steps summarize the primary process of developing the models for forecasting short-and mid-term air temperature.

**Table 3. Air temperature input design for Baker and Crary stations.**

| Model | Input groups | Output | Scale | Training data records | Testing data records |
|---|---|---|---|---|---|
| M1 | $T_{t-1}$ | $T_t$ | Daily | 5624 | 2411 |
| M2 | $T_{t-1}, T_{t-3}$ | $T_t$ | Daily | 5622 | 2411 |
| M3 | $T_{t-1}, T_{t-3}, T_{t-4}$ | $T_t$ | Daily | 5621 | 2411 |
| M4 | $T_{t-1}, T_{t-3}, T_{t-4}, T_{t-5}$ | $T_t$ | Daily | 5620 | 2411 |
| M5 | $T_{t-1}, T_{t-3}, T_{t-4}, T_{t-5}, T_{t-6}$ | $T_t$ | Daily | 5619 | 2411 |
| M6 | $T_{t-1}, T_{t-3}, T_{t-4}, T_{t-5}, T_{t-6}, T_{t-7}$ | $T_t$ | Daily | 5618 | 2411 |
| M7 | $T_{t-1}, T_{t-3}, T_{t-4}, T_{t-5}, T_{t-6}, T_{t-7}, T_{t-8}$ | $T_t$ | Daily | 5617 | 2411 |
| M8 | $T_{t-1}, T_{t-3}, T_{t-4}, T_{t-5}, T_{t-6}, T_{t-7}, T_{t-8}, T_{t-9}$ | $T_t$ | Daily | 5616 | 2411 |
| M9 | $T_{t-1}, T_{t-3}, T_{t-4}, T_{t-5}, T_{t-6}, T_{t-7}, T_{t-8}, T_{t-9}, T_{t-10}$ | $T_t$ | Daily | 5615 | 2411 |
| M10 | $T_{Wt-1}$ | $T_{Wt}$ | Weekly | 803 | 344 |
| M11 | $T_{Wt-1}, T_{Wt-2}$ | $T_{Wt}$ | Weekly | 802 | 344 |
| M12 | $T_{Wt-1}, T_{Wt-2}, T_{Wt-3}$ | $T_{Wt}$ | Weekly | 801 | 344 |
| M13 | $T_{Wt-1}, T_{Wt-2}, T_{Wt-3}, T_{Wt-4}$ | $T_{Wt}$ | Weekly | 800 | 344 |
| M14 | $T_{Wt-1}, T_{Wt-2}, T_{Wt-3}, T_{Wt-4}, T_{Wt-5}$ | $T_{Wt}$ | Weekly | 799 | 344 |
| M15 | $T_{Wt-1}, T_{Wt-2}, T_{Wt-3}, T_{Wt-4}, T_{Wt-5}, T_{Wt-6}$ | $T_{Wt}$ | Weekly | 798 | 344 |
| M16 | $T_{Wt-1}, T_{Wt-2}, T_{Wt-3}, T_{Wt-4}, T_{Wt-5}, T_{Wt-6}, T_{Wt-7}$ | $T_{Wt}$ | Weekly | 797 | 344 |
| M17 | $T_{Wt-1}, T_{Wt-2}, T_{Wt-3}, T_{Wt-4}, T_{Wt-5}, T_{Wt-6}, T_{Wt-7}, T_{Wt-8}$ | $T_{Wt}$ | Weekly | 796 | 344 |

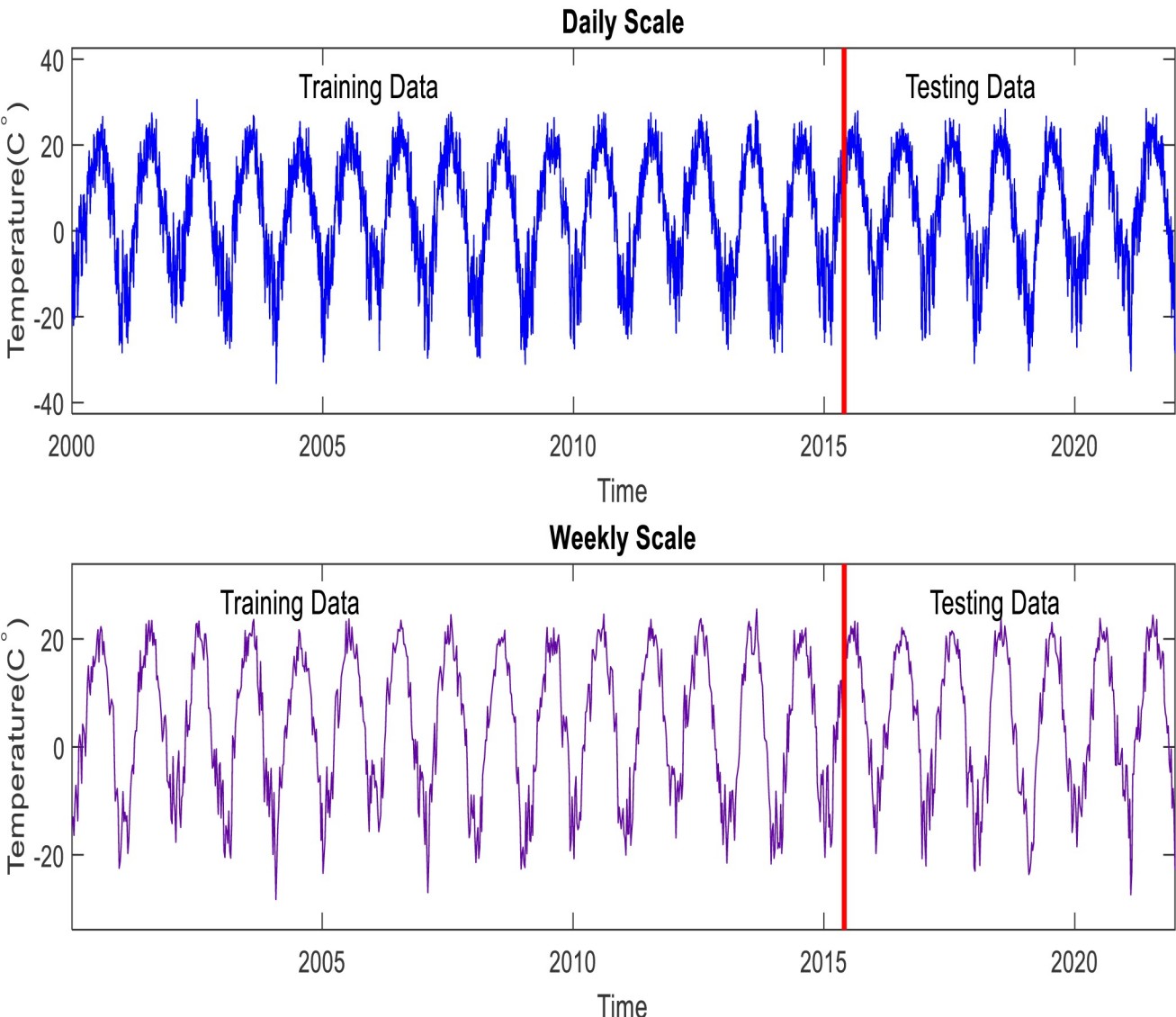

**Fig 3. Daily and hourly air temperature measured in Crary station over the period from 2000 to 2021.**

1. Collecting the daily data for air temperature from a continental climate. In this study, Crary station is selected.

2. Converting the temperature values from Fahrenheit to Celsius using the following formula.

$$T_C = \frac{5}{9}(T_F - 32) \tag{20}$$

$T_C$ and $T_F$ are temperatures measured in degrees Celsius and Fahrenheit.

3. Computing the mean weekly temperatures.

4. Selecting the best lags using ACF, and PACF for both scales (weekly and daily).

5. Data partition: The time-series data is relatively long (2000 to 2021), 70% of the data are used for model calibration (2000 to 2015), and the rest are used for testing. The number of data points was fixed in that case to ensure a fair evaluation of the proposed model throughout the most critical step (testing phase). Notably, this procedure does not affect the data partitions. For example, for daily scale 2411, which represents about 30% of the entire daily records (Table 3)

6. Normalizing the training and testing dataset based on the minimum and maximum temperature in the training data set using the following formula [56]:

$$T_{normalized} = \frac{T_{i-}T_{min}}{T_{max-}T_{min}} \tag{21}$$

$T_{normalized}$ is the normalized temperature for $i^{th}$ temperature record ($T_i$) while $T_{min}$ and $T_{max}$ are minimum and maximum and temperature data obtained from the training data set.

7. Assigning the hyperparameters of the applied models. The trial- and error- method is used for this process where each model was trained 100 times over the training dataset with different parameters. When these models were trained several times, the best ones were selected according to the statistical criteria. According to several statistical metrics, the model which generates lowest forecasting error in the training step is selected. In addition, the performance of the model should be stable so that there is no significant difference between its performance in the training and testing phase.

8. De-normalizing the data based on the following formula:

$$T_{i=}T_{min} + \left( T_{noralize_i} \left( T_{max-}T_{min} \right) \right) \tag{22}$$

9. Evaluating the accuracy of the models with the testing dataset.

It is important to mention that all models are constructed using MATLAB software. The candidate parameters of the applied models can be illustrated below:

• RF: The number of trees is selected between 20–100 while the leaf node ranges from 1–5.

• GBR: the learning rate ∈ [0 1] and the number of trees ∈ [150,1]. In Bag fraction = 1.

• SVR: Box Constraints "regularization parameter "is set between 0.7 to 1. mean the sigma ranges from 0.8452 to 0.7071.

• Epsilon parameter ∈ [0.6 1] and sigma. Finally, the kernel scale parameter ranges from 0.8 to 1.

• DT: MaxNumSplits (maximum number of decision splits) ∈ [1 8]. Tree depth controllers ∈ [5 10].

• Bag fraction = 1 for assembling models that's mean "roughly 2/3 of input data is selected for training for every tree and the remaining 1/3 is used as out-of-bag observations".

In the second scenario of this work, we used the randomization method to divide the data into the training and testing phase. In this scenario, the effect of the climate would be more

obvious on the performance of the AI models. Accordingly, the models in this scenario will be trained using some recent records.

## 2.8 Statistical metrics

Different statistical metrics assess the best model accuracy in daily and weekly temperature forecasting. Furthermore, it is vital to recognize the most efficient model with the least forecasted error. Four statistical measures can be adopted to examine the forecasting accuracy of the suggested modeling approaches, such as root mean square error (RMSE), correlation coefficient (R), Thiels' U-statistics (U), and mean absolute error (MAE). The mathematical expressions of these measures are presented below [57, 58].

$$\text{RMSE} = \sqrt{\frac{1}{N}\sum_{i=1}^{N}\left(A_i - P_i\right)^2} \tag{23}$$

$$\text{MAE} = \sqrt{\frac{1}{N}\sum_{i=1}^{N}\left|A_i - P_i\right|} \tag{24}$$

$$U = \frac{\sum_{i=1}^{N}\left(A_i - P_i\right)^2}{\sqrt{\sum_{i=1}^{N}A_i{}^2} + \sqrt{\sum_{i=1}^{N}P_i{}^2}} \tag{25}$$

$$R = \frac{\sum_{i=1}^{N}\left(A_i - A^-\right)\left(A_i - P^-\right)}{\sqrt{\sum_{i=1}^{N}\left(A_i - A^-\right)^2}\sqrt{\sum_{i=1}^{N}\left(P_i - P^-\right)^2}} \tag{26}$$

Where, $A_i$, and $P_i$ are the actual and forecasting temperature for $i^{th}$ observation. While $A^-$, and $P^-$ are the mean of actual and predicted value, and N is the total number of observations. The stated statistical parameters have been used frequently in the literature for model comparison, and their estimation can be achieved directly from the observed and predicted values. Based on the results of these statistical measures, the model presenting the lowest value of forecasting error and the highest value of R (close to one) is selected as the best model for predicting the air temperature for short- and mid-term forecasting.

## 3. Result and discussion

### 3.1 First scenario

This scenario investigates the capability of the AI models to predict the one-step ahead values for daily and weekly temperature. In this part of the work, we divide the data into two phases: training and testing. The classical method is used to separate the data into two steps; the first 70% of the recorded temperature is used for training, and the last 30% of the data is used for testing. In this scenario, the effect of the current temperature trends is not considered. In other words, the current records of temperatures are used in the testing phase. Thus, the models are tested and evaluated based on their ability to predict the current temperature values of the time series data. Furthermore, the input lags were determined by AFC and produced to the adopted models like RF, SVR, GBR, RT, and QRT. Different statistical parameters and comparable figures are used to assess the models' performances.

   This part discusses the performance of the proposed models for predicting the daily and weekly temperature over the training and testing phases for different input lags. Tables 4 and 5 show the performance of the proposed models during the training phase for both daily and

**Table 4. The performance of the proposed models for daily temperature prediction: Training phase.**

| Models | GBR | | | | QRT | | | | RT | | | | SVR | | | | RF | | | |
|---|---|---|---|---|---|---|---|---|---|---|---|---|---|---|---|---|---|---|---|---|
| | RMSE | R | U | MAE | RMSE | R | U | MAE | RMSE | R | U | MAE | RMSE | R | U | MAE | RMSE | R | U | MAE |
| M1 | 3.663 | 0.963 | 0.132 | 2.829 | 3.183 | 0.972 | 0.113 | 2.24 | 3.09 | 0.974 | 0.111 | 2.374 | 3.789 | 0.96 | 0.135 | 2.897 | **1.807** | **0.991** | **0.065** | **1.389** |
| M2 | 3.573 | 0.965 | 0.128 | 2.771 | 2.763 | 0.979 | 0.099 | 1.878 | 2.851 | 0.978 | 0.102 | 2.171 | 3.745 | 0.961 | 0.134 | 2.886 | 2.824 | 0.978 | 0.101 | 2.157 |
| M3 | 3.553 | 0.965 | 0.128 | 2.759 | 2.621 | 0.981 | 0.094 | 1.768 | 2.75 | 0.979 | 0.099 | 2.123 | 3.734 | 0.961 | 0.134 | 2.873 | 2.756 | 0.979 | 0.099 | 2.117 |
| M4 | 3.521 | 0.966 | 0.127 | 2.734 | 2.357 | 0.985 | 0.084 | 1.562 | 2.529 | 0.983 | 0.091 | 1.923 | 3.724 | 0.962 | 0.133 | 2.864 | 2.521 | 0.983 | 0.090 | 1.919 |
| **M5** | **3.423** | **0.968** | **0.123** | **2.666** | 2.281 | 0.986 | 0.081 | 1.51 | 2.486 | 0.983 | 0.089 | 1.896 | 3.716 | 0.962 | 0.133 | 2.861 | 2.488 | 0.983 | 0.089 | 1.895 |
| M6 | 3.472 | 0.967 | 0.125 | 2.701 | 2.267 | 0.986 | 0.081 | 1.49 | 2.488 | 0.983 | 0.089 | 1.892 | 3.709 | 0.962 | 0.133 | 2.853 | 2.497 | 0.983 | 0.090 | 1.904 |
| M7 | 3.488 | 0.967 | 0.126 | 2.72 | 2.109 | 0.988 | 0.075 | 1.37 | 2.4 | 0.984 | 0.086 | 1.817 | 3.698 | 0.962 | 0.133 | 2.851 | 2.366 | 0.985 | 0.085 | 1.792 |
| **M8** | 3.639 | 0.965 | 0.133 | 2.875 | 2.094 | 0.988 | 0.075 | 1.361 | **2.369** | **0.985** | **0.085** | **1.795** | 3.692 | 0.962 | 0.132 | 2.841 | 2.368 | 0.985 | 0.085 | 1.800 |
| **M9** | 3.503 | 0.966 | 0.126 | 2.734 | **2.087** | **0.988** | **0.074** | **1.349** | 2.387 | 0.985 | 0.086 | 1.812 | **3.682** | **0.963** | **0.132** | **2.837** | 2.369 | 0.985 | 0.085 | 1.803 |

weekly temperature prediction. For daily forecast, all models provide satisfactory results. Nevertheless, the RF provides the best performance, with RMSE ranging from 1.807 to 2.824, R ranging from 0.978 to 0.991, U ranging from 0.065 to 0.101, and MAE ranging from 1.389 to 2.157, followed by the QRT model with RMSE ranging from 2.0874 to 3.1830, R ranging from 0.9722 to 0.9882, U ranging from 0.0745 to 0.1134, and MAE ranging from 1.3492 to 2.24, and RT model with RMSE ranging from 2.3687 to 3.09, R ranging from 0.9738 to 0.9849, U ranging from 0.0849 to 0.1106, and MAE ranging from 1.7951 to 2.374. While GBR and SVR models came last with RMSE ranging from 3.4233 to 3.6633 and from 3.6821 to 3.7887, R ranging from 0.963 to 0.9677 and from 0.9604 to 0.9626, U ranging from 0.1229 to 0.1328 and from 0.1321 to 0.1353, and MAE ranging from 2.6659 to 2.8746 and from 2.8369 to 2.8965, respectively. Furthermore, increasing the number of input lags increases the accuracy of the QRT, RT, and SVR models. Here the QRT model reaches the optimum accuracy when the input lag is nine (QRT – M9), the RT model with eight input lags (RT – M8), SVR model with ten input lags (SVR – M9) and the GBR model with five input lags (GBR – M5). On the contrary, the RF model only requires one input lag to reach its optimum performance RF–M1.

For weekly temperature prediction, all models provide satisfactory predictions reaching the best performance with the QRT model with RMSE ranging from 2.5069 to 3.7157, R ranging from 0.9589 to 0.9817, U ranging from 0.0932 to 0.11386, and MAE ranging from 1.6300 to 2.5880, followed by RF model with RMSE ranging from 2.799 to 3.647, R ranging from 0.960 to 0.977, U ranging from 0.105 to 0.136, and MAE ranging from 2.116 to 2.787. Moreover, increasing the input lags for weekly prediction improves the accuracy of QRT, RF, RT, and SVR models which allows them to reach their optimum accuracy (QRT – M16, RF – M16,

**Table 5. The performance of the proposed models for weekly temperature prediction: Training phase.**

| Models | GBR | | | | QRT | | | | RT | | | | SVR | | | | RF | | | |
|---|---|---|---|---|---|---|---|---|---|---|---|---|---|---|---|---|---|---|---|---|
| | RMSE | R | U | MAE | RMSE | R | U | MAE | RMSE | R | U | MAE | RMSE | R | U | MAE | RMSE | R | U | MAE |
| M10 | 3.898 | 0.954 | 0.146 | 3.02 | 3.716 | 0.959 | 0.139 | 2.588 | 3.663 | 0.96 | 0.137 | 2.792 | 4.648 | 0.935 | 0.174 | 3.607 | 3.647 | 0.960 | 0.136 | 2.787 |
| M11 | 3.797 | 0.957 | 0.143 | 2.971 | 3.486 | 0.964 | 0.13 | 2.379 | 3.522 | 0.963 | 0.132 | 2.713 | 4.607 | 0.936 | 0.173 | 3.584 | 3.458 | 0.965 | 0.130 | 2.669 |
| M12 | 3.837 | 0.956 | 0.144 | 2.992 | 3.256 | 0.969 | 0.121 | 2.215 | 3.578 | 0.962 | 0.135 | 2.72 | 4.598 | 0.936 | 0.173 | 3.572 | 3.455 | 0.965 | 0.130 | 2.668 |
| M13 | 3.333 | 0.967 | 0.125 | 2.605 | 2.898 | 0.975 | 0.108 | 1.93 | 3.167 | 0.971 | 0.119 | 2.398 | 4.543 | 0.938 | 0.17 | 3.528 | 3.126 | 0.972 | 0.117 | 2.400 |
| **M14** | **3.125** | **0.971** | **0.117** | **2.428** | 2.756 | 0.978 | 0.103 | 1.827 | 3.036 | 0.973 | 0.114 | 2.332 | 4.473 | 0.94 | 0.167 | 3.459 | 3.075 | 0.973 | 0.115 | 2.341 |
| M15 | 3.466 | 0.965 | 0.131 | 2.72 | 2.645 | 0.98 | 0.099 | 1.737 | 3.019 | 0.974 | 0.113 | 2.302 | 4.422 | 0.941 | 0.166 | 3.383 | 3.036 | 0.973 | 0.114 | 2.305 |
| **M16** | 3.392 | 0.967 | 0.128 | 2.651 | **2.507** | **0.982** | **0.093** | **1.63** | **2.816** | **0.977** | **0.105** | **2.142** | **4.308** | **0.944** | **0.162** | **3.303** | **2.799** | **0.977** | **0.105** | **2.116** |
| **M17** | 3.827 | 0.962 | 0.147 | 3.044 | 2.853 | 0.976 | 0.106 | 1.880 | 2.822 | 0.977 | 0.106 | 2.144 | 3.532 | 0.963 | 0.131 | 2.689 | 2.737 | 0.978 | 0.102 | 2.109 |

**Table 6. The performance of the proposed models for daily temperature prediction: Testing phase.**

| Models | GBR | | | | QRT | | | | RT | | | | SVR | | | | RF* | | | |
|---|---|---|---|---|---|---|---|---|---|---|---|---|---|---|---|---|---|---|---|---|
| | RMSE | R | U | MAE | RMSE | R | U | MAE | RMSE | R | U | MAE | RMSE | R | U | MAE | RMSE | R | U | MAE |
| M1 | 3.697 | 0.961 | 0.131 | 2.84 | 4.015 | 0.954 | 0.141 | 3.07 | 3.978 | 0.955 | 0.14 | 3.055 | 3.66 | 0.962 | 0.129 | 2.79 | **1.776** | **0.991** | **0.063** | **1.353** |
| M2 | 3.654 | 0.962 | 0.129 | 2.808 | 3.808 | 0.959 | 0.134 | 2.913 | 3.748 | 0.96 | 0.132 | 2.883 | 3.616 | 0.963 | 0.127 | 2.77 | 3.765 | 0.960 | 0.133 | 2.898 |
| M3 | 3.639 | 0.963 | 0.129 | 2.807 | 3.735 | 0.96 | 0.132 | 2.866 | 3.725 | 0.961 | 0.132 | 2.872 | 3.606 | 0.963 | 0.127 | 2.758 | 3.740 | 0.960 | 0.132 | 2.884 |
| M4 | 3.634 | 0.963 | 0.129 | 2.795 | 3.744 | 0.96 | 0.132 | 2.866 | 3.693 | 0.961 | 0.131 | 2.849 | 3.602 | 0.963 | 0.127 | 2.75 | 3.687 | 0.961 | 0.130 | 2.841 |
| **M5** | 3.623 | 0.963 | 0.128 | 2.794 | 3.701 | 0.961 | 0.13 | 2.839 | **3.67** | **0.962** | **0.13** | **2.831** | **3.592** | **0.964** | **0.127** | **2.745** | 3.678 | 0.962 | 0.130 | 2.836 |
| **M6** | **3.612** | **0.963** | **0.128** | **2.791** | 3.697 | 0.961 | 0.13 | 2.851 | 3.691 | 0.961 | 0.131 | 2.848 | 3.593 | 0.963 | 0.127 | 2.747 | 3.686 | 0.962 | 0.130 | 2.841 |
| M7 | 3.617 | 0.963 | 0.128 | 2.791 | 3.714 | 0.961 | 0.131 | 2.842 | 3.68 | 0.962 | 0.13 | 2.855 | 3.593 | 0.963 | 0.127 | 2.755 | 3.696 | 0.961 | 0.131 | 2.850 |
| M8 | 3.714 | 0.963 | 0.134 | 2.917 | 3.727 | 0.961 | 0.131 | 2.843 | 3.684 | 0.962 | 0.13 | 2.838 | 3.593 | 0.963 | 0.127 | 2.752 | 3.690 | 0.961 | 0.131 | 2.849 |
| **M9** | 3.633 | 0.963 | 0.129 | 2.813 | **3.691** | **0.961** | **0.13** | **2.835** | 3.692 | 0.961 | 0.131 | 2.857 | 3.595 | 0.963 | 0.127 | 2.758 | 3.720 | 0.961 | 0.132 | 2.872 |

* Symbol is the hyperparameter for the best model are number of trees = 4, and leaf node = 6.

RT − M16, SVR − M16). It is observed that any increase in the number of lags beyond a value of seven imparts a negative effect on the model performance. At the same time, the GBR model requires five lags (same as the daily prediction) to reach its optimum accuracy (GBR − M14). Overall, QRT, RF, RT, and SVR models better predict daily temperature than weekly during the training phase. On the other hand, the GBR model better predicts the weekly temperature.

Based on the training phase, all models perform very well in predicting the daily and weekly temperature. However, the assessment of the model performances based on the testing dataset is also crucial. For the training phase, the models are provided with complete data (input and targets), which may result in overfitting. Thus, excluding the models' performances in the testing phase may provide users with misleading results. It is known that in the testing phase models received only input features and thus the forecasting accuracy would be more reliable than in the training phase [46, 59].

During the testing phase, the performance of the proposed models was assessed firstly by comparing the performance with each other and secondly by comparing the performance with the ARIMA model as a benchmark model. Table 6 shows the performance of the proposed models during the testing phase for daily temperature prediction. For daily forecast, the RF model provides the best performance, with RMSE ranging from 1.776 to 3.765, R ranging from 0.960 to 0.991, U ranging from 0.063 to 0.133, and MAE ranging from 1.353 to 2.898 followed by the SVR model with RMSE ranging from 3.5915 to 3.6599, R ranging from 0.9621 to 0.9635, U ranging from 0.1265 to 0.1288, and MAE ranging from 2.7451 to 2.7902. Moreover, the RF model requires only one input lag (RF − M1) to reach the best accuracy, while the SVR model requires five input lags. On the other hand, despite the RT and QRT models showing the best performance during the training phase, they came last during the testing phase as they have a tendency to overfit during the training phase.

For further assessment, the ARIMA model was implemented for daily and weekly predictions using two different scenarios. The first one, the ARIMA used for the prediction of temperature using a raw data set. However, the second one, data preprocessing, is used to improve the capacity of ARIMA. At that stage, the differencing method is used to remove seasonality. It is possible to utilize that method to get rid of the temporal reliance, also known as the series dependence on time. The best prediction results are used as a benchmark to validate the AI models. Itis important to mention that the time series data became smoother after the application of differencing transformation technique (see Fig 4a and 4b). Considering the PACF

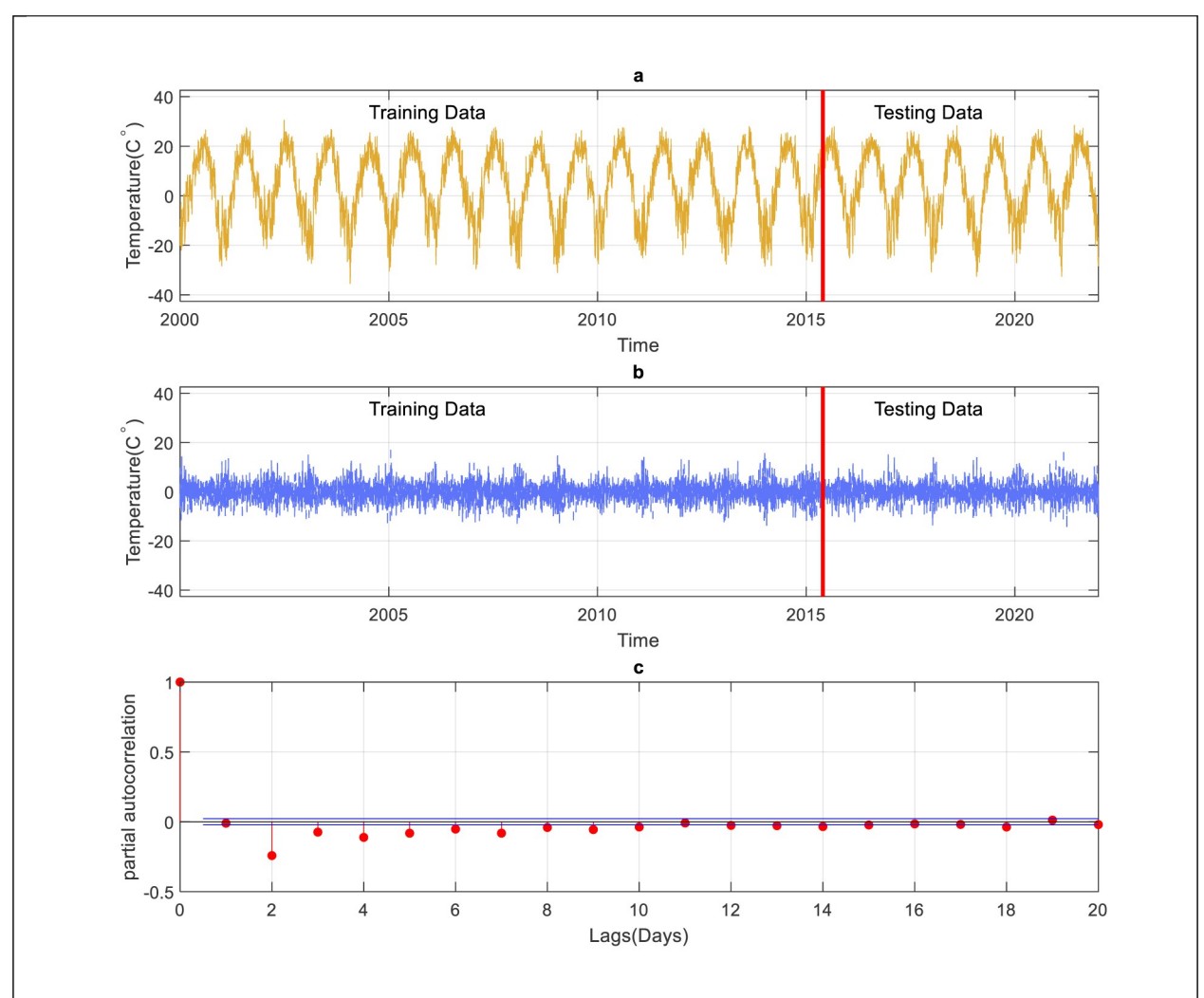

**Fig 4. Input determination for ARIMA model, a) is the original data, b) after applying differencing method, c) PACF for daily scale.**

presented in Fig 4c for daily temperature prediction, three input lags (2,3, and 4 days) are considered for the model development. As shown in Table 7, the ARIMA model performs on par with the QRT model and underperforms in comparison to the other models for daily scale. Furthermore, the ARIMA model requires only three input lags (ARIMA 3,1,2) to reach its best performance. On top of that, the application of the data transformation approach enhances the efficiency of the ARIMA prediction considerably. On the other hand, compared to the ARIMA model, the RF improves the prediction accuracy by 3% in terms of R and reduces the prediction error by 52.33%, 51.16%, and 52.1% in terms of RMSE, U, and MAE, respectively.

For weekly temperature prediction, as presented in Table 8, the RF model demonstrates excellent performance in weekly temperature prediction by providing high prediction accuracy with R ranging from 0.933 to 0.982 and fewer prediction errors with RMSE ranging from 2.478 to 4.665, U ranging from 0.091 to 0.173 and MAE ranging from 1.874 to 3.614 compared to the other models. Furthermore, the high performance of the RF model was achieved using only one input lag (RF − M10), while the other models require significantly higher input lags (seven lags) to reach their optimum performance (SVR − M16, RT − M16, GBR − M16,

**Table 7. The performance of the ARIMA model for daily temperature prediction: Testing phase.**

| | Without removing | | | | With removing | | | |
|---|---|---|---|---|---|---|---|---|
| Model | RMSE | R | U | MAE | RMSE | R | U | MAE |
| ARIMA (2,1,1) | 21.868 | -0.053 | 0.596 | 17.418 | 3.727 | 0.961 | 0.13 | 2.828 |
| ARIMA (2,2,1) | 169.046 | -0.034 | 0.903 | 142.136 | 205.266 | 0.115 | 0.949 | 156.224 |
| ARIMA (2,1,2) | 21.489 | -0.053 | 0.594 | 17.034 | 3.738 | 0.961 | 0.129 | 2.825 |
| ARIMA (3,1,1) | 21.526 | -0.053 | 0.594 | 17.073 | 3.727 | 0.961 | 0.129 | 2.826 |
| ARIMA (3,1,2) | **21.493** | **-0.053** | **0.594** | **17.038** | **3.726** | **0.961** | **0.129** | **2.825** |
| ARIMA (4,1,2) | 21.714 | -0.053 | 0.595 | 17.261 | 3.786 | 0.961 | 0.13 | 2.859 |
| ARIMA (4,1,1) | 21.361 | -0.053 | 0.593 | 16.91 | 3.786 | 0.961 | 0.13 | 2.859 |

**Table 8. The performance of the proposed models for weekly temperature prediction: Testing phase.**

| Models | GBR | | | | QRT | | | | RT | | | | SVR | | | | RF* | | | |
|---|---|---|---|---|---|---|---|---|---|---|---|---|---|---|---|---|---|---|---|---|
| | RMSE | R | U | MAE | RMSE | R | U | MAE | RMSE | R | U | MAE | RMSE | R | U | MAE | RMSE | R | U | MAE |
| **M10** | 4.693 | 0.932 | 0.172 | 3.62 | 4.922 | 0.925 | 0.179 | 3.795 | 4.796 | 0.929 | 0.175 | 3.721 | 4.54 | 0.936 | 0.167 | 3.465 | **2.478** | **0.982** | **0.091** | **1.874** |
| M11 | 4.611 | 0.934 | 0.17 | 3.556 | 4.703 | 0.932 | 0.172 | 3.632 | 4.615 | 0.934 | 0.169 | 3.561 | 4.51 | 0.937 | 0.166 | 3.476 | 4.580 | 0.935 | 0.168 | 3.516 |
| M12 | 4.575 | 0.935 | 0.168 | 3.475 | 4.73 | 0.931 | 0.174 | 3.628 | 4.654 | 0.933 | 0.172 | 3.588 | 4.5 | 0.938 | 0.166 | 3.463 | 4.665 | 0.933 | 0.173 | 3.614 |
| M13 | 4.618 | 0.934 | 0.17 | 3.5 | 4.649 | 0.933 | 0.17 | 3.556 | 4.605 | 0.934 | 0.17 | 3.576 | 4.461 | 0.939 | 0.164 | 3.44 | 4.597 | 0.935 | 0.169 | 3.564 |
| M14 | 4.557 | 0.936 | 0.167 | 3.513 | 4.653 | 0.933 | 0.171 | 3.564 | 4.585 | 0.935 | 0.17 | 3.525 | 4.416 | 0.94 | 0.162 | 3.404 | 4.562 | 0.936 | 0.169 | 3.525 |
| M15 | 4.443 | 0.94 | 0.165 | 3.494 | 4.487 | 0.938 | 0.165 | 3.404 | 4.441 | 0.939 | 0.165 | 3.406 | 4.365 | 0.941 | 0.161 | 3.348 | 4.492 | 0.938 | 0.167 | 3.506 |
| **M16** | **4.423** | **0.941** | **0.165** | **3.472** | **4.423** | **0.94** | **0.163** | **3.382** | **4.247** | **0.945** | **0.157** | **3.291** | **4.286** | **0.944** | **0.158** | **3.283** | 4.358 | 0.941 | 0.161 | 3.349 |
| M17 | 4.582 | 0.941 | 0.174 | 3.707 | 4.552 | 0.936 | 0.168 | 3.521 | 4.297 | 0.943 | 0.160 | 3.262 | 4.613 | 0.934 | 0.169 | 3.495 | 4.293 | 0.943 | 0.159 | 3.305 |

* Symbol is the hyperparameter for the best model are number of trees = 50, and leaf node = 5.

QRT – M16) and increasing the lags beyond seven tends to reduce the models' performance. Overall, the proposed models performed slightly better in predicting the daily temperatures than the weekly ones.

On the other hand, the performance of the ARIMA model for weekly temperature prediction is presented in Table 9. Notably, the differencing method smoothens the time series data by removing a seasonal signal from a series (see Fig 5a and 5b). According to Fig 5c, based on PACF, three lags (1, 2, and 3 weeks) have been considered for the model development. As shown in Table 9, the performance of the ARIMA model is significantly lower than the RF model, and the latter has been able to increase the prediction accuracy by 5.36% in terms of R and reduce the prediction error by 48%, 47%, and 48% in terms of RMSE, U, and MAE.

**Table 9. The performance of the ARIMA model for weekly temperature prediction: Testing phase.**

| | Without removing | | | | With removing | | | |
|---|---|---|---|---|---|---|---|---|
| Model | RMSE | R | U | MAE | RMSE | R | U | MAE |
| ARIMA (1,1,1) | 16.566 | -0.058 | 0.567 | 12.983 | **4.77** | **0.932** | **0.172** | **3.622** |
| ARIMA (1,1,2) | 16.144 | -0.058 | 0.565 | 12.711 | 4.797 | 0.932 | 0.171 | 3.657 |
| ARIMA (2,1,1) | 16.506 | -0.058 | 0.567 | 12.94 | 4.84 | 0.932 | 0.172 | 3.701 |
| ARIMA (2,2,1) | 196.082 | -0.061 | 0.917 | 171.482 | 38.946 | 0.488 | 0.763 | 31.783 |
| ARIMA (2,1,2) | 21.698 | -0.047 | 0.596 | 17.36 | 4.797 | 0.932 | 0.171 | 3.657 |
| ARIMA (3,1,1) | 18.169 | -0.086 | 0.573 | 14.07 | 4.847 | 0.932 | 0.172 | 3.708 |
| ARIMA (3,1,2) | 22.362 | -0.042 | 0.602 | 18.035 | 4.794 | 0.932 | 0.171 | 3.652 |

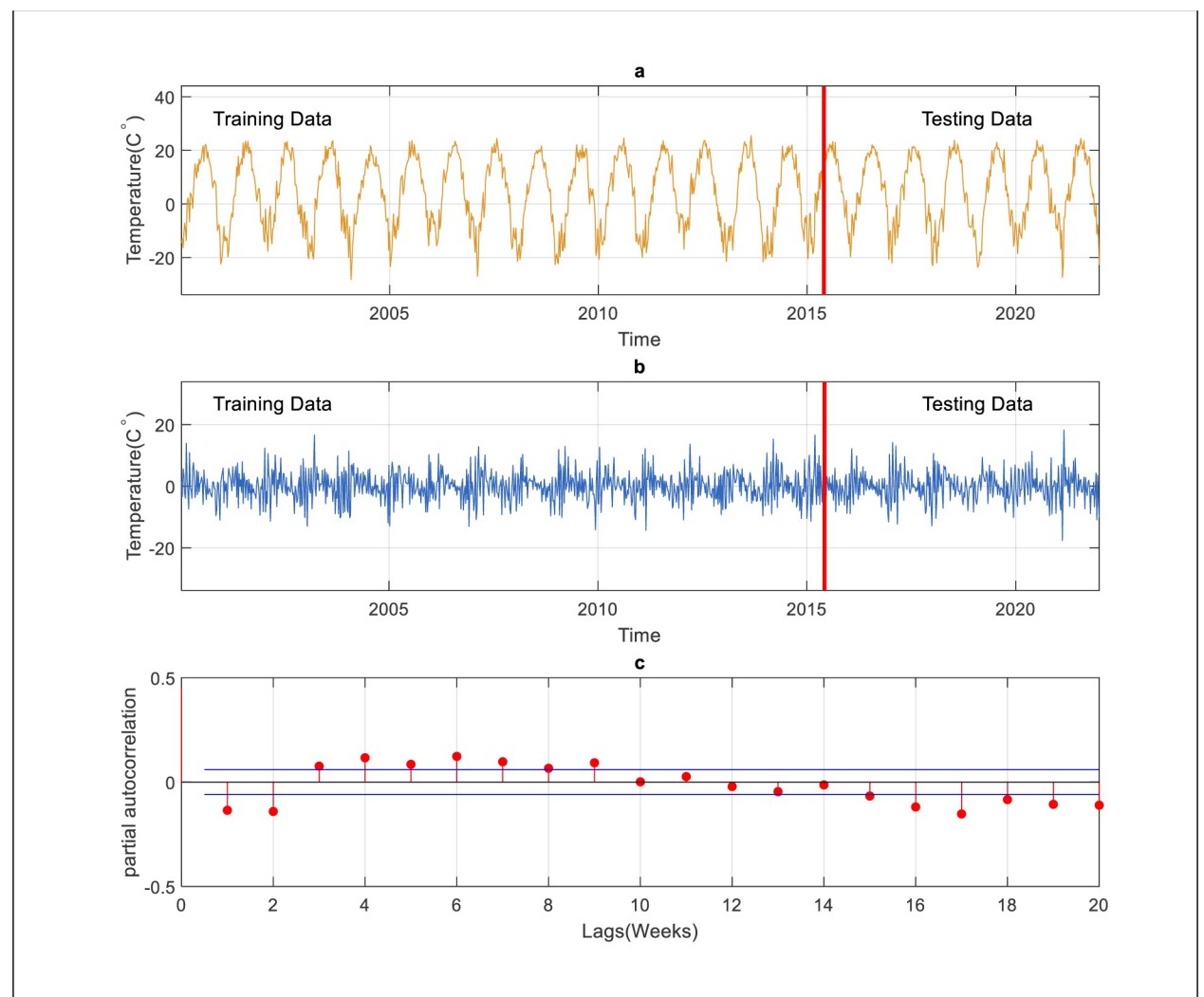

**Fig 5.** Input determination for ARIMA model., a) is the original data, b) after applying differencing method, c) PACF for weekly scale.

Furthermore, the ARIMA model requires only one input lag (ARIMA 1,1,1) for weekly prediction to reach its best performance.

For further assessment, we compared the performance of the best models obtained from the daily prediction (SVR − M5, RT − M5, GBR − M6, QRT − M9, and RF-M1) for each month of the year. In other words, the daily temperature prediction may vary from month to month, so it is essential to investigate the performance of the applied models for each month. What supports the importance of conducting this investigation is the considerable variation in temperature during the months of the year (see Table 2). It can be observed that the Standard deviation (St.D) varies from 2.916 to 7.905. The other significant indicator is that the data length varies monthly (see Fig 6).

Fig 7 shows the performance of the best models based on RMSE statistics for each month of the year. After the training process was completed, the performance of each model was assessed individually. In general, statistical metrics such as RMSE provide the model's overall evaluation. Therefore, this figure is created to see the monthly performance of each model. It is observed that the models have faced problems in estimating the temperatures for the winter

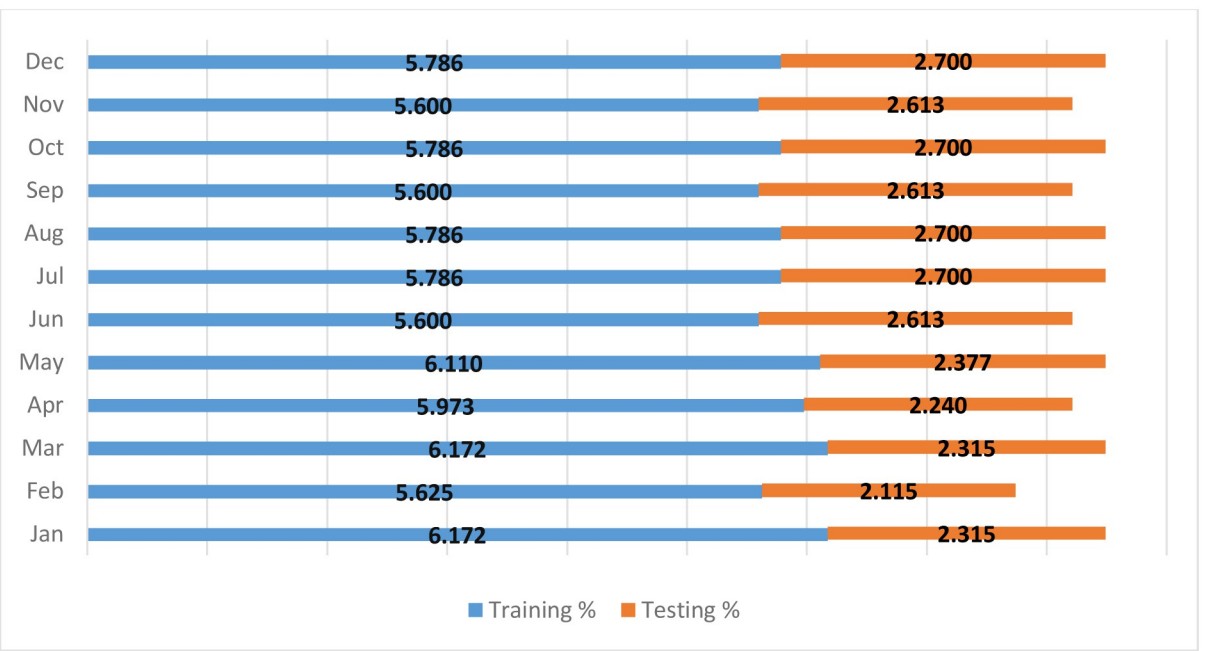

**Fig 6. Percentage of data length used in this study.**

season. According to the Fig 7, the RF and SVR model provided the least amount of prediction error for almost all months, followed by the GBR model. It can be observed that the highest forecasted error is observed in January, February and December. Two reasons may efficiently explain this problem. The first reason may be associated with variability of temperature in the

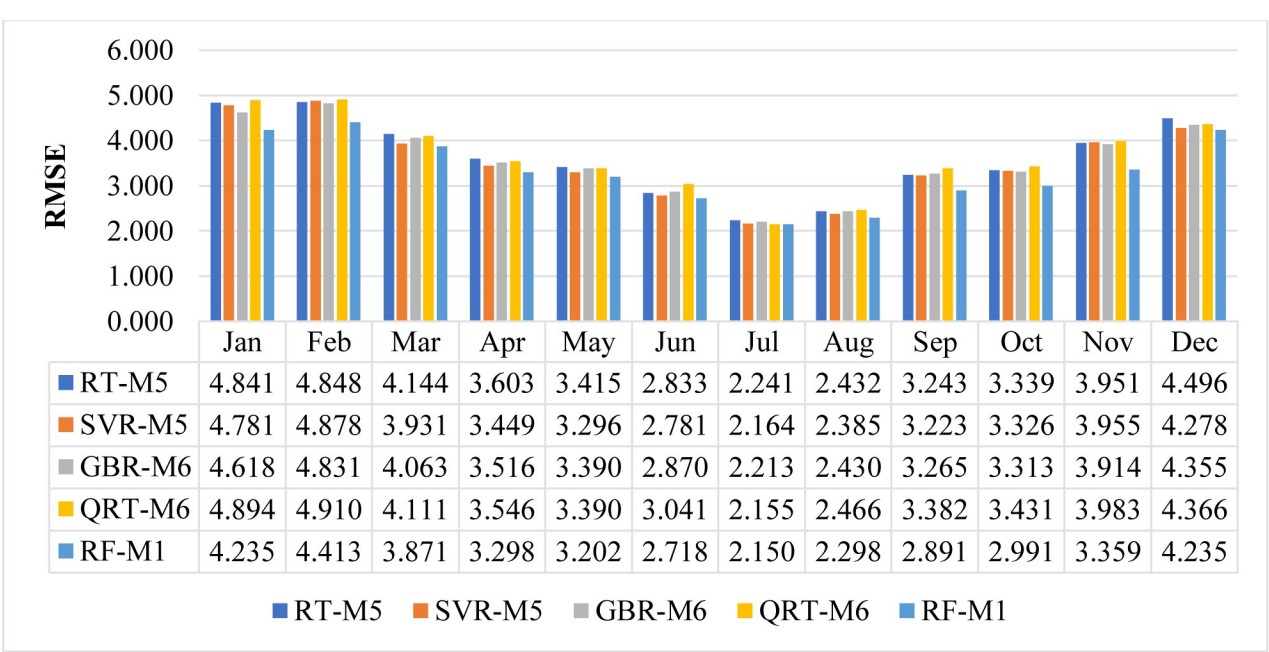

| | Jan | Feb | Mar | Apr | May | Jun | Jul | Aug | Sep | Oct | Nov | Dec |
|---|---|---|---|---|---|---|---|---|---|---|---|---|
| RT-M5 | 4.841 | 4.848 | 4.144 | 3.603 | 3.415 | 2.833 | 2.241 | 2.432 | 3.243 | 3.339 | 3.951 | 4.496 |
| SVR-M5 | 4.781 | 4.878 | 3.931 | 3.449 | 3.296 | 2.781 | 2.164 | 2.385 | 3.223 | 3.326 | 3.955 | 4.278 |
| GBR-M6 | 4.618 | 4.831 | 4.063 | 3.516 | 3.390 | 2.870 | 2.213 | 2.430 | 3.265 | 3.313 | 3.914 | 4.355 |
| QRT-M6 | 4.894 | 4.910 | 4.111 | 3.546 | 3.390 | 3.041 | 2.155 | 2.466 | 3.382 | 3.431 | 3.983 | 4.366 |
| RF-M1 | 4.235 | 4.413 | 3.871 | 3.298 | 3.202 | 2.718 | 2.150 | 2.298 | 2.891 | 2.991 | 3.359 | 4.235 |

**Fig 7. The performance of proposed models in predicting the air temperature for each month.**

winter season (St. D range from 2.916 to 7.905) which leads to a considerable effect on the model performance. The second significant reason is that the training data doesn't have large number of negative extreme values which limits the training efficiency of the model in this scenario.

In terms of the total number of days, February is the shortest month. It can be observed from Fig 7 that the number of data records used in this month constitutes only 7.74% of the total data, which is undoubtedly the lowest percentage of data used in this study. Therefore, the models do not have enough training to simulate that period of the year in which the temperature changes significantly within a short period. Furthermore, the RF provides better efficiency in predicting the temperatures measured in February. The model presents less variance in comparison to the other models. A further noticeable observation related to daily temperature prediction is the fact that all the models except the QRT model require fewer input lags to reach their optimum performance (RF − M1, SVR − M5, RT − M5, GBR − M6), while the QRT model requires more input lags (nine lags) to achieve the optimum performance (QRT − M9).

The performance of the proposed models during the testing phase is also assessed using scatter diagrams (see Figs 8 and 9), histograms (see Figs 10 and 11), and box plots (see Figs 12 and 13). Figs 8 and 9 represent the scatter plot between the observed and predicted temperatures for daily and weekly prediction. The plots examine the cause-effect relationship between the predicted and the observed temperatures and check the degree of association between these two variables in terms of coefficient of determination ($R^2$). For daily prediction, the RF model yielded the best prediction performance in terms of $R^2 \approx 0.983$, while the other models provided slightly similar performance in terms of $R^2$. Additionally, for all data samples, there is considerably less diversion with the ideal line for the RF model compared to the other models. For weekly prediction, the RF model still demonstrated a robust prediction performance with a significantly higher $R^2$ values ($R^2 \approx 0.964$) compared to the other models. At the same time, the RF model showed the least diversion with the ideal line for all data samples compared to the other models.

Figs 10 and 11 show the histogram plots for the forecasting error in the case of both horizons (i.e., daily and weekly) during the testing phase. The plot visually interprets the error distribution by showing the number of error values within a specified range and includes the Gaussian kernel density function to check the error normality. From Figs 10 and 11, it can be inferred that the RF model performs better than the other models in terms of mean error and standard deviation for daily temperature and weekly predictions and provides an error distribution similar to the normal distribution. Moreover, box plots are also constructed to depict the distribution and skewness of forecasting error values by displaying quartiles and averages. The plots display the values in a standardized manner using a five-number summary (i.e., minimum, first quartile, median, third quartile, and maximum) and present more visual information regarding the effectiveness of each model separately. The figures help to better understand the characteristics of forecasting errors generated by the applied models. For daily scale, all models provide the same outlier values, slightly less for the RF − M1 model (see Fig 12a). The quantile of the measured errors is provided in Fig 12b. Accordingly, the RF − M1 model generates a lower interquartile range (IQR = 3.93) than the other models, indicating the efficiency of predicting the daily temperature. For the weekly time scale, the RF − M1 model shows the best performance because its median and mean values are very close to zero compared to other models (Fig 13a). Besides, the generated outliers are fewer than those reported in other models. The most important note can be observed in Fig 13b which shows that the RF-M1 model generates significantly fewer outliers (IQR = 2.554) in comparison to the other models whose IQR ranges from 4.738 to 5.353.

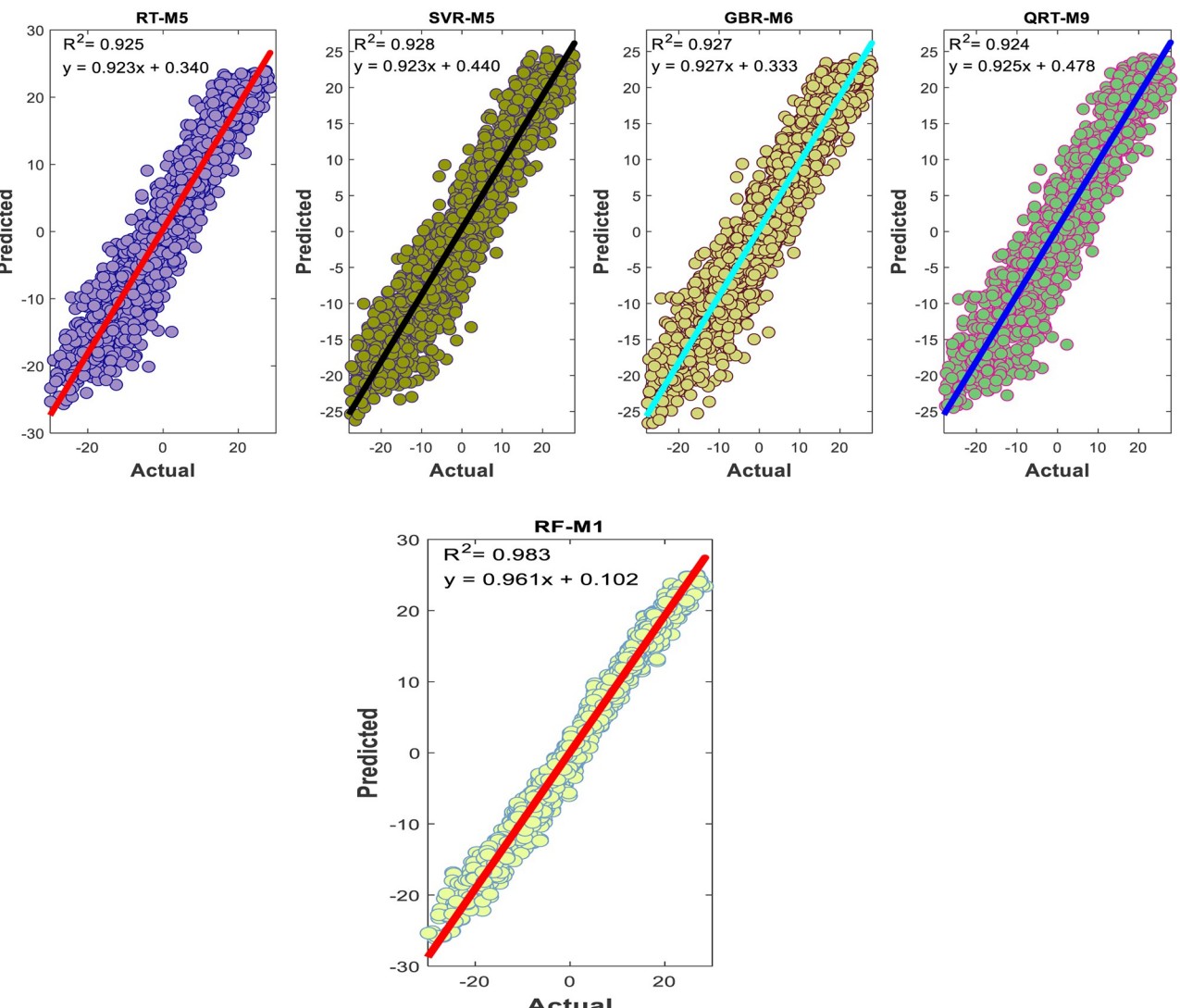

**Fig 8. Comparison between measured daily temperatures and predicted ones through the testing phase.**

For further evaluation, the residual error diagrams for both daily and weekly scales have been developed (Figs 14 and 15). The diagram acts as a performance measure for the applied models and represents the difference between the forecasted and the actual temperature values. It is observed from Figs 14 and 15 that the RF model demonstrates the least residual error in comparison to all other applied models and outmatches them in terms of prediction accuracy and performance.

Lastly, the capacity of the predictive models has been investigated through the hottest months (June, July, and August). These months have the highest temperatures; thus, it is vital to see which applied AI models mimic the extreme temperature values. For that, the probability records (data points), which have lined at 95% confidence interval (mean ± standard deviation), have been computed; it can be seen from Fig 16 that only the RF-M1 model managed to generate more excellent performance than the comparable models. Moreover, the SVR-M5 model could not deal well with the high-temperature values in the hottest months for this study area.

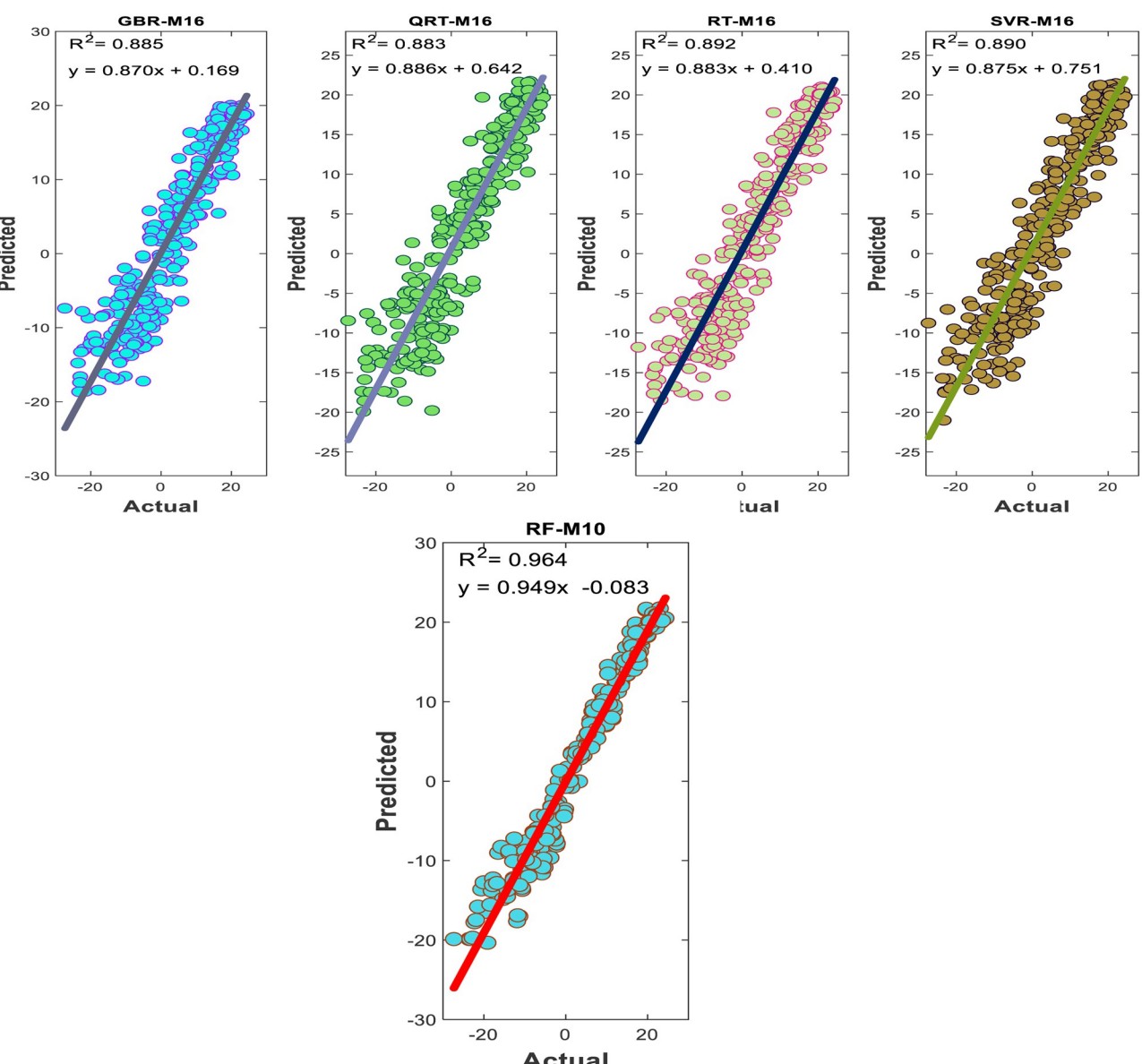

**Fig 9. Comparison between measured weekly temperatures and predicted ones through the testing phase.**

## 3.2 Second scenario

The previously described findings were obtained when data from 2000 to 2015 were utilized for training the applied models. This data counted for 70% of the total records and the rest of the measured data, which represented 30% of the entire data points, was used for testing the models. This type of data division helps to test how these models can simulate the pattern of data recorded in recent years. It is known that the world, in not a few parts of it, is facing a global warming crisis and the time series of temperatures studied in the last decade have shown a behavior and pattern which differs somewhat from what they were observed in the previous years. Accordingly, this study investigates how climate change affects the temperature records. The current records of temperature were used in both the training and testing phases. To do that, the data records are randomly divided into two phases: training (70%) and testing

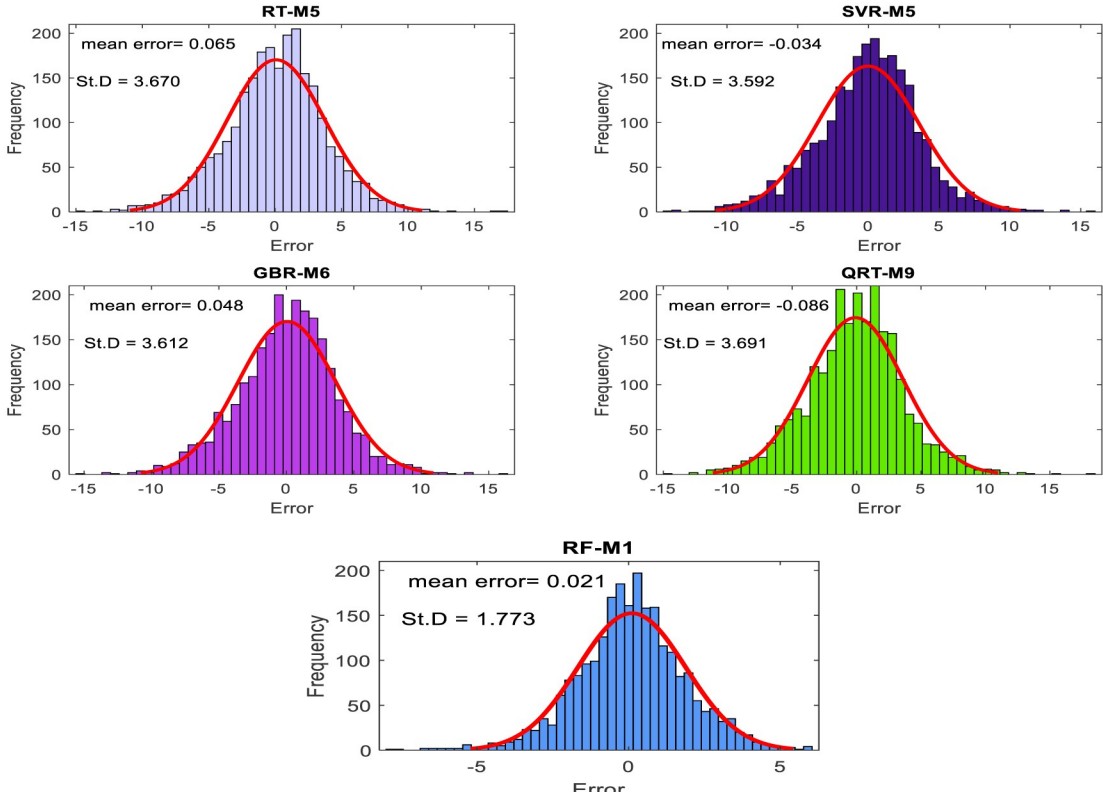

**Fig 10. The histogram and Gaussian kernel density function for daily temperature prediction during the testing phase.**

(30%). After that, the data-driven models (QRT, GBR, RF, and RF) were trained and assessed using several statistical metrics based on this data division. Furthermore, the outcomes of these models are compared with their corresponding results, which have already been discussed. This technique may help observe whether there is an effect on the behavior and accuracy of the model predictions when using recent time series data during the training process. According to the results shown in Table 10, all the predictive models' performances are positively affected using the Randomization method approach. For example, when classical data division procedure was used the RF model generates relatively higher errors (RMSE = 1.776, U = 0.063, and MAE = 1.353) and however, the prediction accuracy is slightly enhanced and the model provides lower forecasting error (RMSE = 1.697, U = 0.061, and MAE = 1.325). Overall the Randomization method has a role in improving the model capacity because it includes features related to future temperature trends in the training data used to train the suggested models in this work.

## 4. Conclusion

The accuracy of the Data-Driven Models, namely RT, SVR, QRT, RF, ARIMA, and GBR, have been investigated to forecast atmospheric air temperature on different time scales (daily and weekly) using historical meteorological data. The data was collected from Cray station, located in North Dakota, USA. This region experiences a volatile continental climate throughout the year. The time-series data is relatively long (2000 to 2021), 70% of the data are used for model calibration (2000 to 2015), and the rest are used for testing. Several input groups were

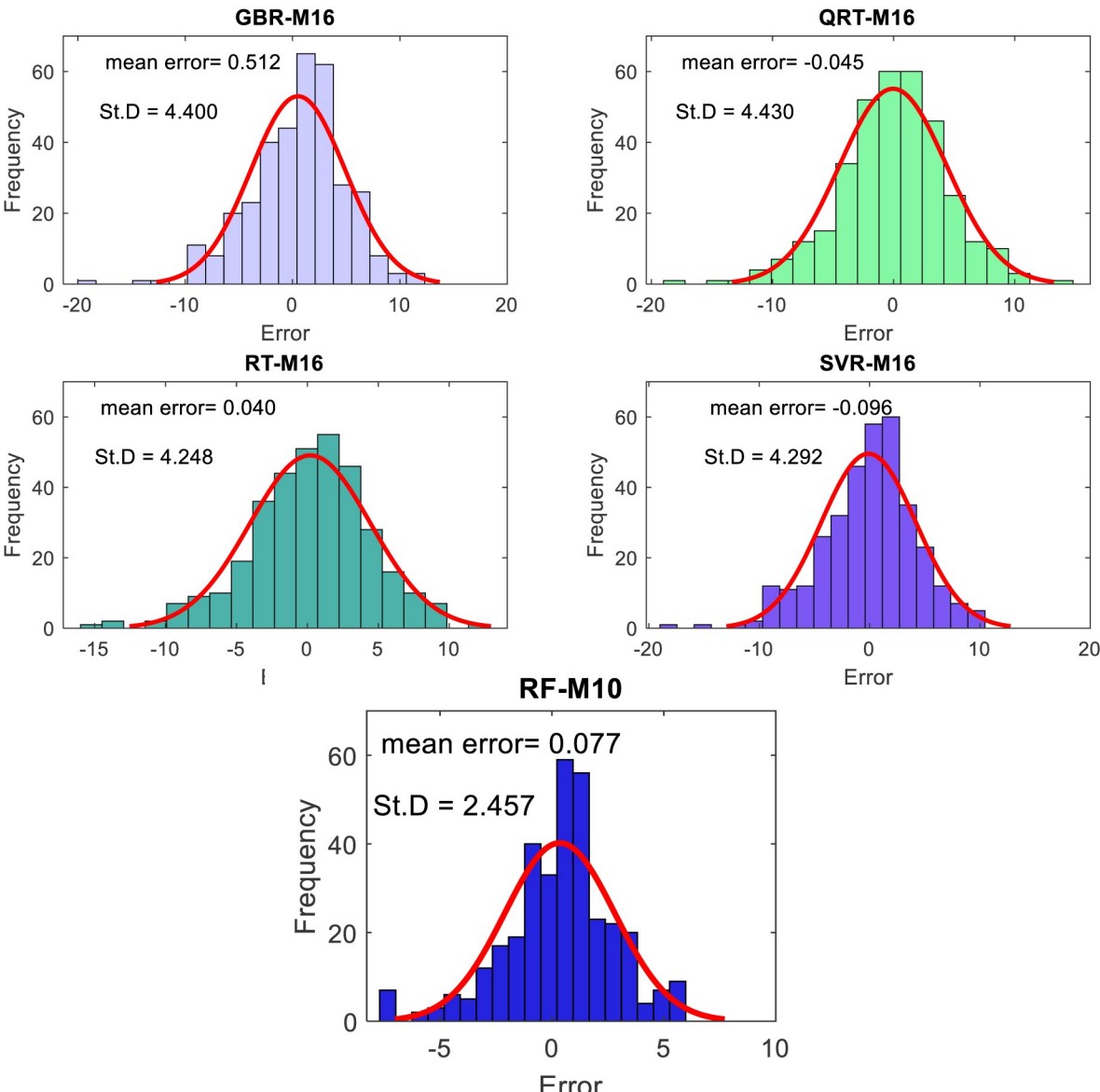

**Fig 11. The histogram and Gaussian kernel density function for weekly temperature prediction during the testing phase.**

examined with different times of lags. The daily scale showed that the RF technique provided more accurate outcomes than the comparable models.

Moreover, the advanced analysis of forecasting error exhibited that the performance of the models was significantly affected by data variability, consistency, and extreme temperature values. As January, February and December had higher variability of temperature data values, the effect on the model performance was greater for these months. In addition to this, the forecasting errors observed for these months were higher than other months due to the fact that the average temperature observed for these months fell below the overall average temperatures observed for the entire dataset.

The models performed very well for the weekly time scale, but the RF, modeling technique provided more accurate results compared to other models. In general, the accuracy of daily

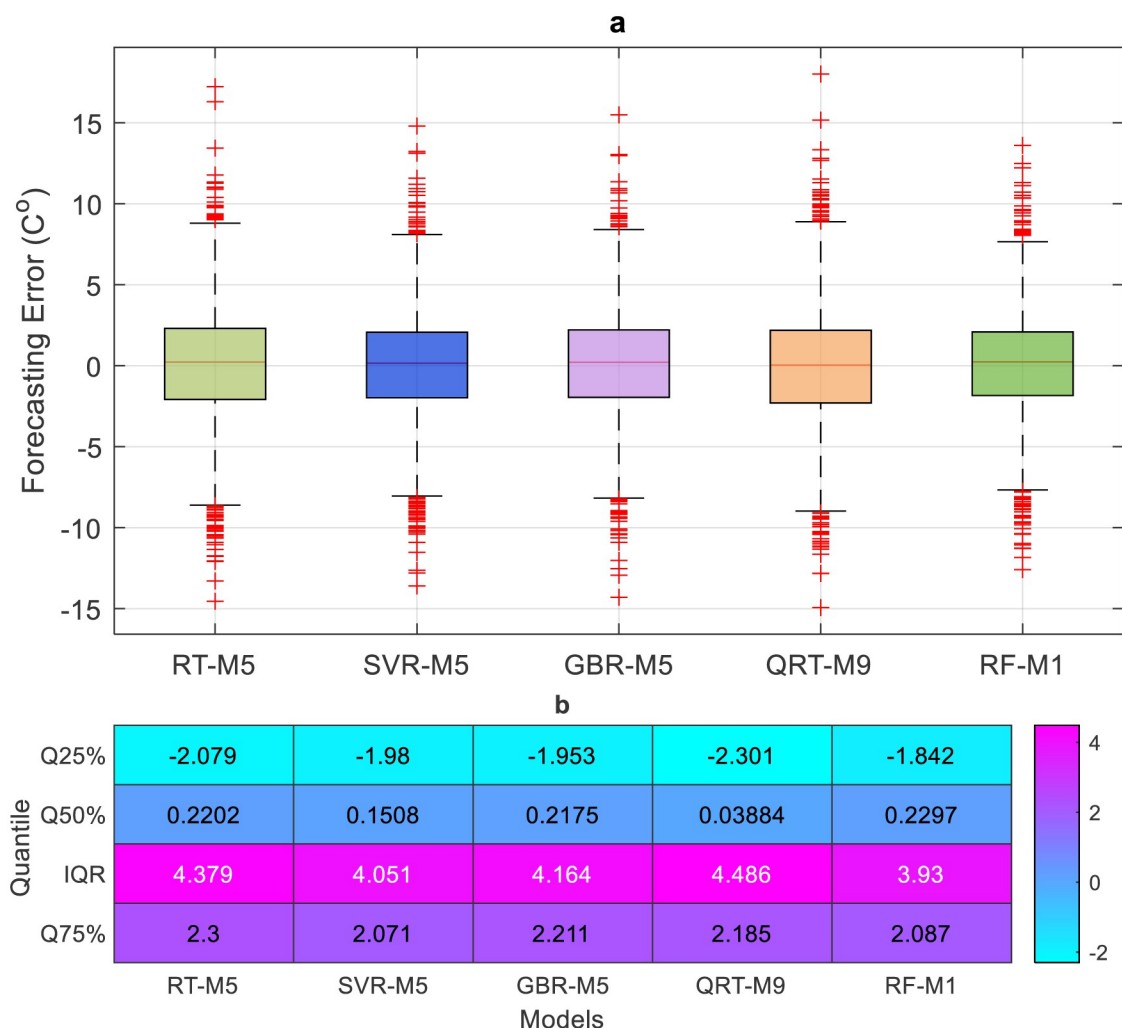

**Fig 12.** a) Boxplot of the forecasting error in daily temperature prediction for all proposed models. b) Quantile percent of the forecasting error.

forecasting temperature was higher than the weekly scale. This may be because the weekly records were calculated by taking the average temperatures for seven days, which led to the loss of some critical data characteristics. Furthermore, as the weekly scale was derived from the daily records, the length of the time-series data reduced significantly, which affected the efficiency of the model during the training process.

This study also investigated the AI models' capacity to predict temperature when the future pattern data is included. In this scenario, the randomization data division was applied to divide the data into training and testing. The study found that the prediction models' performance was enhanced after using these techniques. This means that the current pattern of the temperature data affecting climate change influences the quality of predictions. Besides, the case study location starts to be gradually affected by climate change and its impact on temperature values.

Thus, this study suggests the following recommendations:

- Adopting a robust approach to determine the best input combination instead of existing (ACF and PACF) methods.

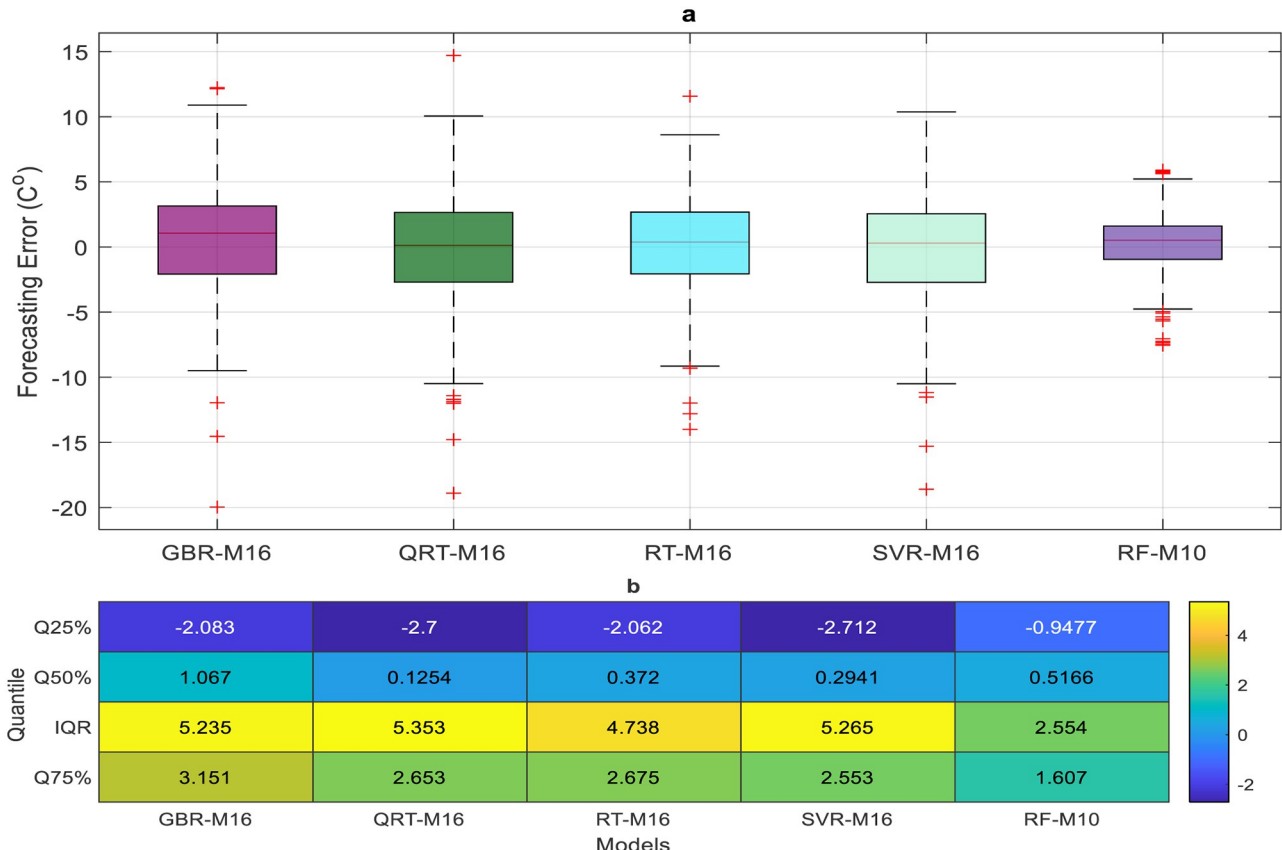

**Fig 13.** a) Boxplot of the forecasting error in weekly temperature prediction for all proposed models. b) Quantile percent of the forecasting error.

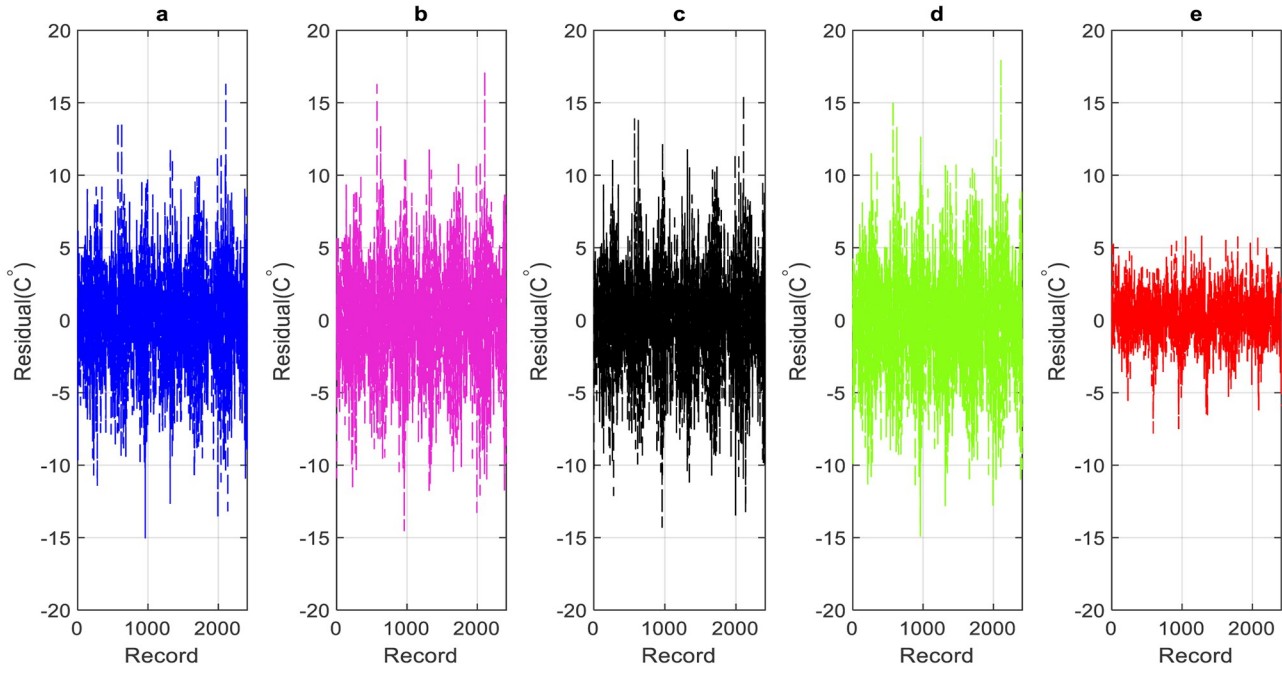

**Fig 14. Daily residual: testing phase.** a) GBR-M6, b) RT-M5. c)SVR-M5. d)QRT-M9. e) RF-M1.

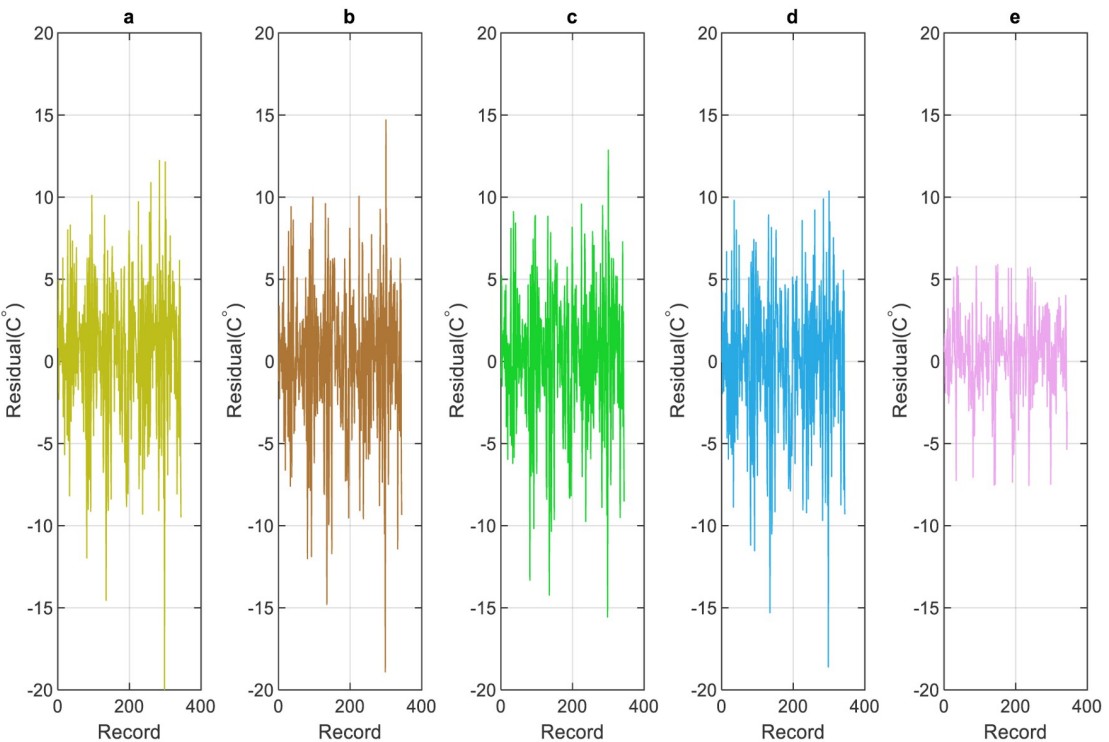

**Fig 15. Weekly residual: testing phase.** a) GBR-M16, b) RT-M16, c) SVR-M16, and e) are RF-M10.

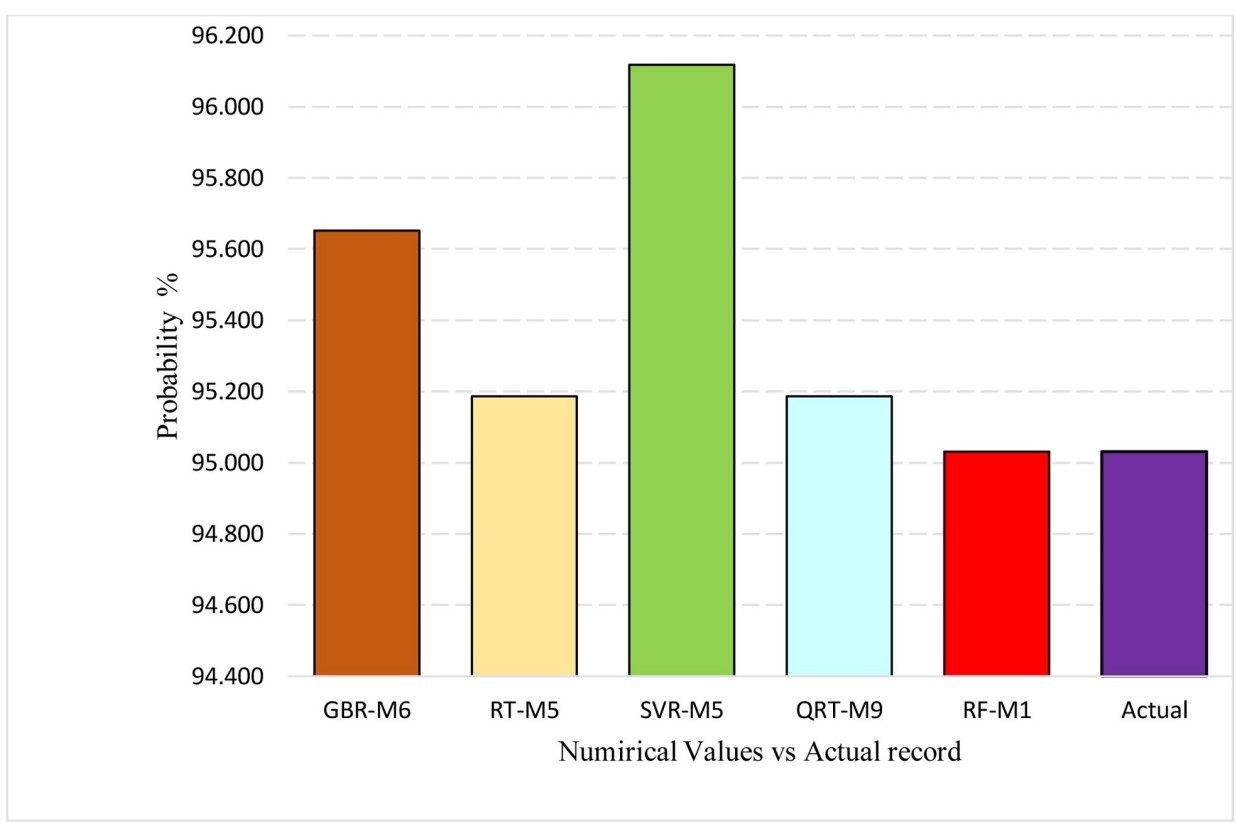

**Fig 16. The probability of the data falling at the confidence level of 95% (μ±2σ).** μ is the average, and σ is the standard deviation.

**Table 10. Comparison of the performance of the models when two different data division methods are applied.**

| | without randomization | | | | Randomization method | | | |
|---|---|---|---|---|---|---|---|---|
| Model | RMSE | R | U | MAE | RMSE | R | U | MAE |
| QRT-M9 | 3.691 | 0.961 | 0.13 | 2.835 | 3.659 | 0.962 | 0.130 | 2.828 |
| GBR-M6 | 3.612 | 0.963 | 0.128 | 2.791 | 3.598 | 0.965 | 0.129 | 2.795 |
| RT-M5 | 3.67 | 0.962 | 0.13 | 2.831 | 3.655 | 0.962 | 0.131 | 2.849 |
| SVR-M5 | 3.592 | 0.964 | 0.127 | 2.745 | 3.582 | 0.963 | 0.129 | 2.760 |
| RF-M1 | 1.776 | 0.991 | 0.063 | 1.353 | 1.697 | 0.992 | 0.061 | 1.325 |

- Applying a Bio-inspirited algorithm to select the optimal hyperparameters of SVR

- Studying to what extent the size of the data used to train the performed models affects the accuracy of predictions. This task can be accomplished using different training and testing rations.

## Acknowledgments

The authors would like to thank the anonymous reviewers for their constructive comments, which significantly improved this research.

## Author Contributions

**Conceptualization:** Adil Masood.

**Data curation:** Mohammed Majeed Hameed.

**Formal analysis:** Faidhalrahman Khaleel, Mohammed Abdulhakim AlSaadi, Mohammed Majeed Hameed.

**Funding acquisition:** Mohammed Abdulhakim AlSaadi.

**Investigation:** Mohamed Khalid Alomar, Siti Fatin Mohd Razali.

**Methodology:** Faidhalrahman Khaleel, Adil Masood, Siti Fatin Mohd Razali.

**Project administration:** Siti Fatin Mohd Razali, Mohammed Abdulhakim AlSaadi.

**Resources:** Siti Fatin Mohd Razali.

**Software:** Mohammed Majeed Hameed.

**Supervision:** Nadhir Al-Ansari.

**Validation:** Faidhalrahman Khaleel.

**Visualization:** Mohammed Majeed Hameed.

**Writing – original draft:** Adil Masood, Mohammed Majeed Hameed.

**Writing – review & editing:** Mohamed Khalid Alomar, Faidhalrahman Khaleel, Mustafa M. Aljumaily.

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
