## [Decision Letter · Decision Letter 0]

22 Jun 2022

PONE-D-22-10602Data-Driven Models for Atmospheric Air Temperature Forecasting at a Continental Climate Region.PLOS ONE

Dear Dr. Hameed,

Thank you for submitting your manuscript to PLOS ONE. After careful consideration, we feel that it has merit but does not fully meet PLOS ONE’s publication criteria as it currently stands. Therefore, we invite you to submit a revised version of the manuscript that addresses the points raised during the review process.

We look forward to receiving your revised manuscript.

Kind regards,

Yangyang Xu

Academic Editor

PLOS ONE

Journal Requirements:

"This study is funded from Almaarif university college"

" The authors like to thank Al-Maarif university college for supporting this study "

"This study is funded from Almaarif university college"

6. We note that Figure 1 in your submission contain [map/satellite] images which may be copyrighted. All PLOS content is published under the Creative Commons Attribution License (CC BY 4.0), which means that the manuscript, images, and Supporting Information files will be freely available online, and any third party is permitted to access, download, copy, distribute, and use these materials in any way, even commercially, with proper attribution. For these reasons, we cannot publish previously copyrighted maps or satellite images created using proprietary data, such as Google software (Google Maps, Street View, and Earth). For more information, see our copyright guidelines: http://journals.plos.org/plosone/s/licenses-and-copyright.

a. You may seek permission from the original copyright holder of Figure(s) [#] to publish the content specifically under the CC BY 4.0 license.  

Reviewers' comments:

Reviewer's Responses to Questions

**Comments to the Author**

1. Is the manuscript technically sound, and do the data support the conclusions?

Reviewer #1: Partly

Reviewer #2: Partly

2. Has the statistical analysis been performed appropriately and rigorously? 

Reviewer #1: No

Reviewer #2: N/A

3. Have the authors made all data underlying the findings in their manuscript fully available?

Reviewer #1: No

Reviewer #2: Yes

4. Is the manuscript presented in an intelligible fashion and written in standard English?

Reviewer #1: Yes

Reviewer #2: Yes

5. Review Comments to the Author

Reviewer #1: The paper presents an interesting case study on applying four machine learning models to forecast air temperature in Cray station of North Dakota using selected historical temperature as features. The forecasting performance of the four models are compared. There are several aspects that are not adequately addressed by the paper (see the major comments below), making the comparison results less solid.

Major comments:

1. As shown in Figure 6. the temperature data in the paper exhibit strong seasonality. A common practice for seasonal time-series data is to decompose the data into a seasonal component and a deseasonalized component, and then use ML models to predict for the latter. This usually improves forecast performance since some of the temporal dependence in the data can be captured by the seasonal models. The paper should also compare with the ML models using deseasonalized data.

2. It is unclear to me how tuning parameters are chosen for each ML model using the "trial- and error- method," and hence the performance comparison in the paper is not entirely convincing (and the results may not be reproducible either). I think the paper should rigorous describe the tuning parameter selection procedure, including but not limited to: What are the hyperparameters being tuned? What are their candidate values? What are the criteria to select the optimum values? What are the selected hyperparameters for the results reported in Section 3?

3. I think more details and justifications are needed on using autocorrelation function (ACF) and partial autocorrelation function (PACF) to choose the number of lags in the candidate models (Table 3). While it seems that the paper is using the lags that lead to PACF values larger than a certain threshold to determine the features included in the daily models, this does not seem to be the case for the weekly models. Also, the weekly models in general achieve better performance by including more features. Does including lag 8 and beyond further improve the performance?

4. The paper should include the performance of traditional time-series models (e.g., ARIMA) as a benchmark, otherwise it is hard to gauge the performance of the four ML models. There are also many off-the-shelf data-based approaches for univariate time-series forecasting, such as THETA [1], Prophet [2] and TBATS [3]. It would be interesting to see what their performance would be for this data set.

5. It is unclear how forecast is performed on the test data set. For example, to forecast $T_t$ on 2020-1-1 using M2, do you plug in the actual values of $T_{t-1}$ and $T_{t-3}$, or do you plug in predicted values of them? From a practical perspective, these two approaches correspond to a one-step-ahead forecast and a multiple-step-ahead forecast, respectively. The paper should clarify which of these two types of forecasting is considered.

6. The paper compares RT and QRT with GBR. I would recommend also comparing their performance with random forest [4] and quantile regression forest [5], which are known to have lower prediction variance than RT and QRT, respectively.

7. In Section 4, the paper summarizes the forecast errors by scatter plots, histograms, and boxplots, which seem redundant to me as they all convey similar information on the model performance. In addition, it would be interesting to see a plot of residuals and/or errors against the time, from which one can examine the temporal patterns of the residuals/errors. A strong temporal dependence in the residuals may suggest that the model is underfitting the dependence among the data and more complex models may be needed.

Minor comments:

1. What is the same size of the training and test data sets in each model?

2. In the second paragraph of Section 2.1, what are the reported summary statistics? The standard deviation for July does not seem correct.

3. In Figure 8, are the models for different months trained by the same data set from 2000-2015, or the model for a particular month is trained by the data from the same month (e.g., the model for January is trained by the January data only)? Due to the strong seasonality in the data, the results from the two approaches may be different.

4. In Equation (1) and (3), there should be a transpose sign between w and x. The optimization formulation in Equation (2) and (3) should include the "slack variables."

5. The mathematical notations should be clearly defined and used consistently. For example, throughout the paper, $\\gamma$, $y$ and $T$ are all used to denote the dependent variable. Equations in Section 2.5 are not correctly numbered.

6. The writing in Section 3 could be improved. For example, the detailed discussion on model fitting during the training phase seems unnecessary and could be misleading as the main focus of the paper is on out-of-sample forecasting.

Reference:

[1] Assimakopoulos, V. and Nikolopoulos, K. (2000). The theta model: a decomposition approach to forecasting. International Journal of Forecasting 16, 521-530.

[2] Taylor SJ, Letham B. (2017). Forecasting at scale. PeerJ Preprints 5:e3190v2 https://doi.org/10.7287/peerj.preprints.3190v2

[3] De Livera, A.M., Hyndman, R.J., & Snyder, R. D. (2011). Forecasting time series with complex seasonal patterns using exponential smoothing, Journal of the American Statistical Association, 106(496), 1513-1527.

[4] Breiman, L. (2001). Random forests. Machine learning, 45(1), 5-32.

[5] Meinshausen, N., & Ridgeway, G. (2006). Quantile regression forests. Journal of Machine Learning Research, 7(6).

Reviewer #2: Review of: Data-Driven Models for Atmospheric Air Temperature Forecasting at a Continental Climate Region.

This paper uses multiple machine-learning or deep-learning methods to predict daily and weekly air temperature, given the air temperature of previous days (or weeks). Since machine learning and deep learning became one of the main techniques that could be used in various fields, application of these methods in weather and atmospheric science has been growing. In that sense, I think this paper could be one of the case studies of application of ML and DL methods in weather prediction.

However, there is some concerns regarding the performance of the model. I think the performance of the developed model is not sufficient to claim that the authors are actually making the prediction of air temperature. Additional analysis to increase the model performance is needed for this paper to be considered for publication.

Major comments:

1. Section 2.2 – 2.4

These methodologies are widely used, and it is unnecessary to explain the standard procedures for each method. What is important is to highlight the advantage of each model and variation of each model that the authors made to be more suitable for this study. For example:

1) the authors used RBF kernel for SVR. What is the bandwidth of the kernel? Is there a regularization parameter?

2) DT methods, what is the depths of the trees? What are the number of samples required to split?

3) Elaborate the pruning method. How is this done?

4) GBR, what is the learning rate? What is the depth of the model?

Overall, there are so many hyperparameters that authors selected, but none of these values were given. It will be more informative to give this number than to explain the standard model background. Furthermore, with the values of hyperparameters given, it is essential to perform a sensitivity test on the hyperparameters – authors note that trial and error with 100 iteration was used for this. Was there a big difference in the performance of the model? What are the selected parameters?

2. Please elaborate the ACF and PACF (such as equations). Why are these functions used to determine the best lags?

3. On the other hand, despite the RT and QRT models showing the best performance during the training phase, they came last during the testing phase since they may overfit the data during the training phase.

Did authors fix the hyperparameters to prevent the overfitting? I think if the authors guess that overfitting happened, they should fix the models.

4. For the daily predictions, MAE of the best model (SVR) is about 2.7degC, and for weekly predictions, MAE is 3.3degC. The problem with these numbers is that we don’t know how good these values are. So, there should be some kind of baseline to compare to see how good these results are. For example, what would be the MAE of daily predictions if we just use the previous day’s temperature as prediction? Or, for example, what if we just use the seasonality of temperature for the prediction?

5. One of the concerns is that 30% of the data at the end of the observation is used as testing dataset. As authors mentioned in the introduction, the climate is changing. So, the physical drivers of temperature as well as mean and trend of temperature and change during the observation. The authors can use 30% of randomly selected data as a testing data or add in “year” term to the model and see how the results change.

6. More detailed assessment of the predictions should be made. For example, looking at Fig. 10, the prediction errors are higher in colder temperatures. Naturally this is reasonable since temperature variability is higher in winter. Seasonal analysis of temperature should be included. – Now that I look at Figure 8, I see the seasonal analysis. Please talk about this in the main text, and please provide some insights on increase of error in winter is happening.

7. The authors discussed outliers (or extreme temperature events) using the IQR. One of the most important features in forecasting temperature is ability to capture the extreme temperatures. So, rather than using IQR with all datasets, authors should see how the models capture the seasonal extreme temperatures (for example, 2sigma values in June, July August, for heat extreme).

8. Overall, I don’t think the model is doing a good job on forecasting temperature. I took the referenced data (is it Cray or Crary? – I could not find station named Cray). For the 2000-2021 period, I simply compared daily temperature and the daily temperature the day before, and the RMSE was 3.83 degC for the entire period and MAE was 2.91 degC. I mentioned this in the previous comment, but what this means is that the model is not significantly better than simply taking the temperature of day before as its predictions. Furthermore, mean error with my calculation was -0.0015, and STD was 3.83. Comparing this with Fig 11, there is no significant improvement in the model. Attempt to utilize ML to forecast temperature is indeed valuable, but better model performance is needed for this research to be justified.

Minor comments:

It is well known that numerous meteorological and ecological events, human life, and crops in agricultural areas are significantly influenced by climate conditions as well as the environment's physical conditions.

- “Environment’s physical conditions” are rather vague. Please clarify or give examples.

Given these climate conditions, the temperature is a critical factor that can change and is one of the most significant meteorological parameters.

- What are “these climate conditions”? This sentence is confusing. Are you trying to say that temperature can alter other environments?

Moreover, air temperature is one of the most influential factors in the evapotranspiration phenomenon, which is vital for managing water resources and agricultural activities.

- “phenomenon” can be deleted

Furthermore, it has been observed that the temperature varies significantly, which may be responsible for the changes in weather throughout the time [41]

- This in unclear. Are you referring to seasonal variation of temperature or year-to-year variation of extreme temperatures?

The upper air patterns of each season have distinctive features, bringing several weather conditions.

- What are the distinctive features and several weather conditions? Elaborate or delete this sentence.

Accordingly, the continental climate of North Dakota makes the forecasting of the weather patterns a problematic task.

- Why is it difficult to forecast the weather continental climate compared to other climates? Please elaborate.

Tables 1 and 2 show the statistical characteristics of the minimum, mean, average, standard deviation, and skewness of the daily and weekly air temperature at the Cray meteorological station from 2000 to 2021.

- Is it Cray or Crary? I could not find station named Cray. Please make sure.

Are all the diagrams for the models (Figs 2-4) really necessary? I advise to delete these figures.

There are lots of interesting figures, but it does not get talked in the main text. Please give some elaboration of figures in the main text.

6. PLOS authors have the option to publish the peer review history of their article (what does this mean?). If published, this will include your full peer review and any attached files.

Reviewer #1: No

Reviewer #2: No

---

## [Author Response · Author response to Decision Letter 0]

10 Aug 2022

Dear Editor and reviewers,

All the received comments have been carefully addressed.

---

## [Decision Letter · Decision Letter 1]

4 Sep 2022

PONE-D-22-10602R1Data-Driven Models for Atmospheric Air Temperature Forecasting at a Continental Climate Region.PLOS ONE

Dear Dr. Hameed,

Thank you for submitting your manuscript to PLOS ONE. After careful consideration, we feel that it has merit but does not fully meet PLOS ONE’s publication criteria as it currently stands. Therefore, we invite you to submit a revised version of the manuscript that addresses the points raised during the review process.

We look forward to receiving your revised manuscript.

Kind regards,

Yangyang Xu

Academic Editor

PLOS ONE

Journal Requirements:

Reviewers' comments:

Reviewer's Responses to Questions

**Comments to the Author**

1. If the authors have adequately addressed your comments raised in a previous round of review and you feel that this manuscript is now acceptable for publication, you may indicate that here to bypass the “Comments to the Author” section, enter your conflict of interest statement in the “Confidential to Editor” section, and submit your "Accept" recommendation.

Reviewer #2: (No Response)

2. Is the manuscript technically sound, and do the data support the conclusions?

Reviewer #2: Yes

3. Has the statistical analysis been performed appropriately and rigorously? 

Reviewer #2: Yes

4. Have the authors made all data underlying the findings in their manuscript fully available?

Reviewer #2: Yes

5. Is the manuscript presented in an intelligible fashion and written in standard English?

Reviewer #2: Yes

6. Review Comments to the Author

Reviewer #2: I appreciate the effort authors made to improve the paper. I think this is publishable after minor revisions.

1. This paragraph is unclear

North Dakota is located in the middle of North America and is subjected to extreme climate conditions, with hot summers and cold winters.

- Isn’t this the same for all regions? Why did you select North Dakota?

Furthermore, it has been observed that the temperature varies extremely from season to season, which may be responsible for the changes in weather throughout the time [42].

- Same. Of course, weather is causing temperature variation. Why is ND special? Or does it represent the general climate in continental US?

The continental climate of North Dakota makes forecasting weather patterns a problematic task. The difficulty in forecasting may be related to the climate throughout the year. Knowing that the high variation in temperatures through the seasons may hinder predicting the temperature

- It is confusing what the authors are trying to tell us. Please clarify.

2. On SVR method, authors say: “14 However, this equation is unreliable in many hydrological challenges (nonlinear- regression analyses)”. However, the authors are predicting temperature, which is not hydrological (it might be in some aspect, but not generally). Please clarify or make it consistent.

3. This is clearly wrong: The first reason may be associated with the length of the data set used to develop the models. The St.D of the temperature records in February is very high (St.D = 7.695). The second significant reason is the data length used in this study.

The first reason is associated with variability of temperature in wintertime, not the length of the data. Furthermore, the authors claim this is due to the insufficient data (in second reason). I assume this is because Feb has only 28 days, compared to other months with 30, 31 days. Do authors think 2-day (or about 8%) difference really make a huge difference in prediction performance? This can be tested by shortening other months to 28 days. If authors are going to claim this, it should be tested. However, I think this is not the main driver that causes the performance difference. The performance difference is also shown on weekly analysis, in which Feb has probably the same lengths (4-5 weeks) as other months.

4. The last reason may be related to the extreme negative values. This case study has a slight increase in temperature over time; therefore, there are more days with high temperatures than days with low temperatures.

I don’t think trend in temperature really matters. Authors are claiming that there are very few extreme-low temperature to be trained, and this is mostly due to high variability of temperature in wintertime, which overlaps with the first reason.

5. Same. Please make more high-level analysis on the reason: Moreover, the advanced analysis of forecasting error exhibits that the performance of the models is significantly affected by the length, consistency, and variability of data. As February has a fewer number of days as well as higher variability of recorded data, the monitored error in this month is higher than in other months.

7. PLOS authors have the option to publish the peer review history of their article (what does this mean?). If published, this will include your full peer review and any attached files.

Reviewer #2: No

---

## [Author Response · Author response to Decision Letter 1]

14 Sep 2022

Dear Editor,

Thank you for giving us the opportunity to submit a revised draft of our manuscript titled

"Data-Driven Models for Atmospheric Air Temperature Forecasting at a Continental Climate Region" to PLOS One journal. The authors appreciate the time and effort that you and the reviewers have dedicated to providing your valuable feedback on my manuscript. We are grateful to the reviewers for their insightful comments on our paper. Besides, we have been able to incorporate changes to reflect of the suggestions provided by the reviewers. Notably, the changes in the manuscript have been highlighted. 

Here is a point-by-point response to the reviewers' comments and concerns.

Comments from Reviewer 

Reviewer #2: I appreciate the effort authors made to improve the paper. I think this is publishable after minor revisions

1. This paragraph is unclear

North Dakota is located in the middle of North America and is subjected to extreme climate conditions, with hot summers and cold winters.

- Isn’t this the same for all regions? Why did you select North Dakota?

Reply: 

- Yes, because they are located in a continental climate. One of the characteristics of this climate is that there is a difference between temperatures in summer and winter.

- The reasons of selecting the North Dakota are:

1- The location of the case study is in a climate where there is a significant change in the temperatures throughout the seasons.

2- Data availability. 

3- It has a remarkable location. The distance from North Dakota to Equator and North Pole are almost the same. 

4- All the above points were incorporated in the revised manuscript.

Furthermore, it has been observed that the temperature varies extremely from season to season, which may be responsible for the changes in weather throughout the time [42].

- Same. Of course, weather is causing temperature variation. Why is ND special? Or does it represent the general climate in continental US?

Reply: Due to its far inland location and proximity to both the North Pole and the Equator, which is almost equal, there are noticeable temperature fluctuations. We have incorporated this this point in the revised manuscript. 

The continental climate of North Dakota makes forecasting weather patterns a problematic task. The difficulty in forecasting may be related to the climate throughout the year. Knowing that the high variation in temperatures through the seasons may hinder predicting the temperature

- It is confusing what the authors are trying to tell us. Please clarify.

Reply: Thank you for pointing out this problem. The authors with the best of their efforts have provided more clarity on this sentence as per the recommendations of the reviewer.

2. On SVR method, authors say: “14 However, this equation is unreliable in many hydrological challenges (nonlinear- regression analyses)”. However, the authors are predicting temperature, which is not hydrological (it might be in some aspect, but not generally). Please clarify or make it consistent.

Reply: The authors have now clarified this sentence as per the reviewers’ recommendations. The sentence is now more lucid and consistent for the readers.

3. This is clearly wrong: The first reason may be associated with the length of the data set used to develop the models. The St.D of the temperature records in February is very high (St.D = 7.695). The second significant reason is the data length used in this study. The first reason is associated with variability of temperature in wintertime, not the length of the data. Furthermore, the authors claim this is due to the insufficient data (in second reason). I assume this is because Feb has only 28 days, compared to other months with 30, 31 days. Do authors think 2-day (or about 8%) difference really make a huge difference in prediction performance? This can be tested by shortening other months to 28 days. If authors are going to claim this, it should be tested. However, I think this is not the main driver that causes the performance difference. The performance difference is also shown on weekly analysis, in which Feb has probably the same lengths (4-5 weeks) as other months.

Reply: The authors are very thankful to the reviewer for pointing out this issue. The authors totally agree with the point raised by the reviewer which says that the length of data is not the reason which makes a considerable difference in the prediction error. However, the variation of temperature may be the main reason and accordingly the authors have adjusted the manuscript based on the reviewers’ comments. Furthermore, the authors also have focused on the effects of extreme values on model performance. 

4. The last reason may be related to the extreme negative values. This case study has a slight increase in temperature over time; therefore, there are more days with high temperatures than days with low temperatures. I don’t think trend in temperature really matters. Authors are claiming that there are very few extreme-low temperature to be trained, and this is mostly due to high variability of temperature in wintertime, which overlaps with the first reason.

Reply: 

1. The authors completely agree with the reviewer that the first reason related to data duration does not have a significant effect in the context of model performance. Therefore, as per the reviewer’s suggestions, the authors have removed it from the entire manuscript and have disused this issue concerning the variability of data. 

2. Moreover, the authors want to clarify that the negative extreme values interpret the performance of the model for specific months when temperatures are very low (i.e., January, February, and December). For these months, the negative extreme values are fewer in comparison to other months (see table 1) leading to higher forecasting error (see figure 7). 

5. Same. Please make more high-level analysis on the reason: Moreover, the advanced analysis of forecasting error exhibits that the performance of the models is significantly affected by the length, consistency, and variability of data. As February has a fewer number of days as well as higher variability of recorded data, the monitored error in this month is higher than in other months.

Reply: More high-level analysis is performed accordingly. please see pg 27 , line number 4-9

---

## [Decision Letter · Decision Letter 2]

19 Oct 2022

Data-Driven Models for Atmospheric Air Temperature Forecasting at a Continental Climate Region.

PONE-D-22-10602R2

Dear Dr. Hameed,

We’re pleased to inform you that your manuscript has been judged scientifically suitable for publication and will be formally accepted for publication once it meets all outstanding technical requirements.

Kind regards,

Yangyang Xu

Academic Editor

PLOS ONE

Additional Editor Comments (optional):

Reviewers' comments:

Reviewer's Responses to Questions

**Comments to the Author**

1. If the authors have adequately addressed your comments raised in a previous round of review and you feel that this manuscript is now acceptable for publication, you may indicate that here to bypass the “Comments to the Author” section, enter your conflict of interest statement in the “Confidential to Editor” section, and submit your "Accept" recommendation.

Reviewer #2: All comments have been addressed

2. Is the manuscript technically sound, and do the data support the conclusions?

Reviewer #2: Yes

3. Has the statistical analysis been performed appropriately and rigorously? 

Reviewer #2: Yes

4. Have the authors made all data underlying the findings in their manuscript fully available?

Reviewer #2: Yes

5. Is the manuscript presented in an intelligible fashion and written in standard English?

Reviewer #2: Yes

6. Review Comments to the Author

Reviewer #2: I acknowledge the effort authors has made to reflect my suggestions. I think this paper is now publishable.

However, I am not capable of going through minor grammatical issues, so please make sure your paper is grammatically sound.

7. PLOS authors have the option to publish the peer review history of their article (what does this mean?). If published, this will include your full peer review and any attached files.

Reviewer #2: No

---

## [Editor Report · Acceptance letter]

24 Oct 2022

PONE-D-22-10602R2 

*Data-Driven Models for* Atmospheric Air Temperature Forecasting at a Continental Climate Region. 

Dear Dr. Hameed:

I'm pleased to inform you that your manuscript has been deemed suitable for publication in PLOS ONE. Congratulations! Your manuscript is now with our production department. 

Kind regards, 

on behalf of

Dr. Yangyang Xu 

Academic Editor

PLOS ONE